# ExPO-HM: Learning to Explain-then-Detect for Hateful Meme Detection

**Jingbiao Mei**[1,2]\*, **Mingsheng Sun**[2]†, **Jinghong Chen**[1], **Pengda Qin**[4], **Yuhong Li**[5],
**Da Chen**[3]‡, **Bill Byrne**[1]‡

[1]Department of Engineering, University of Cambridge, [2]Xiaohongshu Inc.,
[3]University of Bath, [4]Tencent Company, China, [5]Alibaba Group

jm2245@cam.ac.uk, sunms5513@gmail.com, jc2124@cam.ac.uk,
derekpdqin@tencent.com, daniel.yuhong@gmail.com,
da.chen@bath.edu, wjb31@cam.ac.uk

## Abstract

Hateful memes have emerged as a particularly challenging form of online abuse, motivating the development of automated detection systems. Most prior approaches rely on direct detection, producing only binary predictions. Such models fail to provide the context and explanations that real-world moderation requires. Recent Explain-then-Detect approaches, using Chain-of-Thought prompting or LMM agents, perform worse than simple SFT baselines, and even advanced post-training methods such as GRPO fail to close the gap. Our analysis identifies two key issues of such systems: important policy-relevant cues such as targets and attack types are not hypothesized by the model as a likely explanation; and the binary reward signal is insufficient to guide reasoning. To address these challenges, we propose ExPO-HM (Explain-then-Detect Policy Optimization for Hateful Memes), inspired by the training and evaluation process of human annotators. ExPO-HM combines SFT warmup, GRPO with curriculum learning, and Conditional Decision Entropy (CDE) as both metric and reward for reasoning quality. Across three hateful meme benchmarks, ExPO-HM achieves state-of-the-art performance on binary detection, fine-grained classification, and reasoning quality, with up to 15% and 17% F1 improvement over the GRPO and DPO baselines, respectively. By moving hateful meme detection from simple binary alarms to explanation-driven detection, ExPO-HM provides accurate, interpretable, and actionable moderation support. Code available at: https://github.com/JingbiaoMei/ExPO-HM

*This paper contains content for demonstration purposes that may be disturbing for some readers.*

## 1 Introduction

The rise of social media has led to a surge in hateful content, notably in the form of memes. This has sparked growing research interest in automated hateful meme detection systems that aim at supporting human moderation (Kiela et al., 2020; Liu et al., 2022; Prakash et al., 2023; Shah et al., 2024). Most prior work focuses on direct detection, which only provides a binary classification as to whether a meme is hateful or benign (Cao et al., 2023; Mei et al., 2024; Su et al., 2025). However, recent studies show that moderators require additional information to improve efficiency (Calabrese et al., 2024), such as what type of attack is present, and why the system considers the meme harmful. Additionally, social media users may also benefit from understanding these explanations of harmfulness.

Interestingly, human annotators are not trained and evaluated on binary judgments; common practice is that they are guided by a detailed moderation policy manual that defines policy violations such as

---

\*Work done during internship at Xiaohongshu Inc
†Equal Contribution
‡Corresponding authors

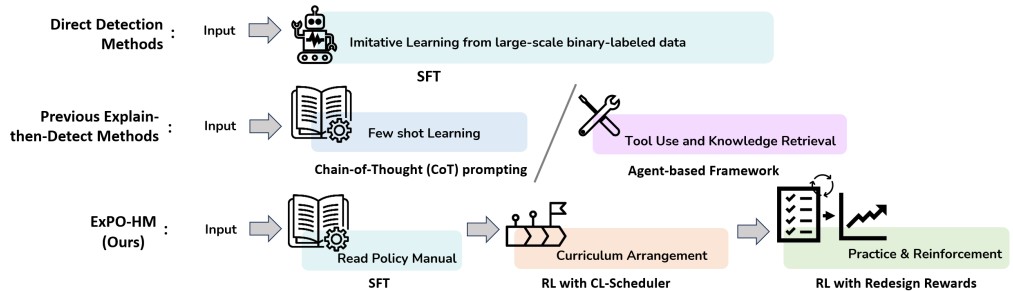

Figure 1: Comparing previous methods with ExPO-HM.

disparagement of protected groups (Singhal et al., 2023). It would be infeasible to train annotators by showing them only raw examples with binary labels; the fine-grained framework provides the necessary structure for both training and evaluation. This human analogy highlights a crucial gap: if humans require fine-grained guidelines and reasoning to make reliable judgments, automated systems could benefit from the same. We call this setting "Explain-then-Detect", where the system first generates a natural language rationale and then produces a classification decision.

Recent work builds Explain-then-Detect Large Multimodal Model (LMM) systems using Chain-of-Thought (CoT) prompting (Wei et al., 2023; Pan et al., 2025) or agent-based frameworks (Huang et al., 2024), but these perform worse than direct Supervised Fine-tuning (SFT) baselines (Mei et al., 2025). Reinforcement learning methods such as Group-Relative Policy Optimization (GRPO) (Shao et al., 2024) can strengthen model reasoning through post-training, yet we find that applying GRPO directly still underperforms SFT for hateful meme detection. Our study reveals two key challenges for Explain-then-Detect systems. First, model explanations often fail to identify the correct violated policy or target, leading to misleading predictions. Second, the binary reward signal in GRPO is too weak to guide reasoning, just as human annotators cannot learn from only yes/no labels.

To address these issues, we propose ExPO-HM (**Ex**plain-then-Detect **P**olicy **O**ptimization for **H**ateful **M**emes), inspired by how human annotators are trained and evaluated. ExPO-HM first uses SFT warmup on a policy manual, mirroring the guideline-based training of human annotators. We then apply GRPO with curriculum learning, mimicking how annotators are first trained and evaluated on fine-grained categories before making binary judgments. We further introduce Conditional Decision Entropy (CDE) both as a metric for explanation quality and as a reward signal to encourage decisive reasoning. We summarize our contributions:

- **Paradigm.** We introduce the first Explain-then-Detect hateful meme detection that outperforms direct detection, enabling accurate and interpretable hateful meme understanding.
- **Methods.** ExPO-HM mimics human moderator training, combining policy manual SFT warmup, GRPO curriculum learning, and CDE-based reward optimization.
- **Evaluation.** We propose a comprehensive evaluation setup that reflects real-world moderation, extending beyond binary classification to fine-grained categories and hateful reasoning judged by LLMs, with extensive baseline comparisons.
- **Results.** ExPO-HM surpasses previous best systems, and achieves new state-of-the-art performance across binary, fine-grained, and reasoning benchmarks, with up to **15%** and **17%** F1 improvement over the GRPO and DPO baseline, respectively.

## 2 RELATED WORK

**Direct Hateful Meme Detection** Most existing approaches to hateful meme detection treat the task as binary classification. Numerous studies fine-tune CLIP-based models using only binary labels and train dedicated classifiers (Pramanick et al., 2021; Kumar & Nandakumar, 2022; Burbi et al., 2023; Cao et al., 2023; Ji et al., 2024; Mei et al., 2024). Decoder-based LMMs have also been fine-tuned for this task (Alayrac et al., 2022; Laurençon et al., 2023; Hu et al., 2024). In particular, Mei et al. (2025) trains a classifier and retriever on top of the LMM embeddings, achieving state-of-the-art binary detection performance.

In contrast, fine-grained classification, such as identifying attack types or target groups, has received far less attention, despite its importance in real-world moderation. Annotated datasets are available (Mathias et al., 2021a; Dimitrov et al., 2021; Fersini et al., 2022; Shah et al., 2024), and some earlier work has explored this problem (Zia et al., 2021; Mathias et al., 2021b), but recent progress has been limited. Mod-Hate (Cao et al., 2024) and IntMeme (Hee & Lee, 2025) leverage fine-grained annotations during training but do not report fine-grained results. MemeCLIP (Shah et al., 2024) addresses this by fine-tuning separate CLIP-based classifiers for each split. In this paper, we systematically evaluate models under different setups and extend the evaluation to fine-grained classification, addressing this important gap.

**Explain-then-Detect Hateful Meme Detection** Compared to direct hateful meme classification, research on explainable hateful meme detection is far more limited. With the rise of decoder-based language models, some Explain-then-Detect systems have emerged. For example, Lin et al. (2024) leverages a debate between two language models to decide meme harmfulness, while LOREHM (Huang et al., 2024) adopts a reasoning-agent framework with retrieval and reflection. However, these systems still primarily target binary classification.

A key challenge is the lack of annotated explanation data. Hatred (Hee et al., 2023), built on the Facebook Hateful Memes dataset (Kiela et al., 2020), remains the only open-source dataset with human-written rationales. Other efforts, such as the recent Arabic hateful meme dataset ArMeme (Kmainasi et al., 2025), are not yet publicly available. Moreover, reasoning tasks remain difficult (Nguyen & Ng, 2024). Existing Explain-then-Detect systems not only struggle with reasoning but also underperform direct detection models in binary classification, underscoring the cost of requiring explanations without tailored optimization strategies. In this paper, we make two key contributions. First, we benchmark a comprehensive set of Explain-then-Detect systems using the Hatred dataset. Second, inspired by human moderator training, we develop ExPO-HM, the first Explain-then-Detect system that surpasses both prior explainable and direct detection approaches, delivering accurate and interpretable hateful meme detection.

## 3 ExPO-HM Methodology

### 3.1 Preliminaries

**Problem Statement.** A common binary hateful memes classification dataset (Kiela et al., 2020) is $\mathcal{D} = \{(I_i, c_i^*)\}_{i=1}^N$, where $I_i \in \mathbb{R}^{C \times H \times W}$ is an image with overlaid text ($C$ for channels, $H$ for height, $W$ for width), and the ground-truth label $c_i^* \in \{0, 1\}$ denotes `benign`/`hateful`. In addition, we consider annotations including fine-grained labels $z_i^*$ (e.g., protected category, attack type) (Mathias et al., 2021a) and, when available, gold explanations (Hee et al., 2023) $\mathbf{e}_i^*$. We thus define the three tasks for hateful meme detection: (1) predicting binary class $c_i$; (2) predicting fine-grained class $z_i$; (3) generating $\mathbf{e}_i$. For text-based evaluation, we denote the textualized label prediction as $d_i$ (from $c_i$ or $z_i$) and the corresponding ground-truth text label as $d_i^*$.

**Large Multimodal Models (LMMs).** Given a meme $I$ and a prompt $p$, we denote the input to LMM as $\mathbf{x} = (I, p)$. An LMM with parameters $\theta$ defines an auto-regressive policy over output text tokens $\mathbf{y} = (y_1, \dots, y_{|\mathbf{y}|})$:

$$\pi_\theta(\mathbf{y} \mid \mathbf{x}) = \prod_{t=1}^{|\mathbf{y}|} \pi_\theta(y_t \mid y_{<t}, \mathbf{x}), \tag{1}$$

where $t$ indexes the output tokens. *Direct-Detection* methods decode labels directly, via answers like "`yes`"/"`no`" (Lin et al., 2024). In contrast, *Explain-then-Detect* first generates reasoning and then the label. Following the standard long CoT format (DeepSeek-AI et al., 2025), the output sequence is:

$$\mathbf{y} \equiv \big(\texttt{<think>} \ \mathbf{e} \ \texttt{</think>} \ \texttt{<answer>} \ d \ \texttt{</answer>}\big), \tag{2}$$

where $\mathbf{e}$ is the generated explanation and $d$ is the textualized label prediction.

**Supervised Fine-Tuning (SFT).** Given an input $\mathbf{x}$ and a target output sequence $\mathbf{y}^*$, the model is trained by maximizing the likelihood of $\mathbf{y}^*$:

$$\mathcal{L}_{\text{SFT}}(\theta) = -\sum_{t=1}^{|\mathbf{y}^*|} \log \pi_\theta(y_t^* \mid \mathbf{y}_{<t}^*, \mathbf{x}). \tag{3}$$

This serves as the general form of SFT used in our baselines.

**Direct Preference Optimization (DPO).** We consider DPO (Rafailov et al., 2023) as a baseline fine-tuning method. Preference pairs $(\mathbf{y}^+, \mathbf{y}^-)$ are sampled on-policy from the reference model $\pi_{\text{ref}}$ via the Explain-then-Detect prompting format. A response $\mathbf{y}$ is selected as the preferred response $\mathbf{y}^+$ if its decision $d$ matches the ground-truth label $d^*$; otherwise, it is treated as the rejected response $\mathbf{y}^-$.

We optimize the DPO objective:

$$\mathcal{L}_{\text{DPO}}(\theta) = -\log \sigma \left( \beta \log \frac{\pi_\theta(\mathbf{y}^+|\mathbf{x})}{\pi_{\text{ref}}(\mathbf{y}^+|\mathbf{x})} - \beta \log \frac{\pi_\theta(\mathbf{y}^-|\mathbf{x})}{\pi_{\text{ref}}(\mathbf{y}^-|\mathbf{x})} \right), \tag{4}$$

where $\sigma$ is the sigmoid function and $\pi_{\text{ref}}$ is the reference model, i.e., the initial model before DPO fine-tuning.

**Group Relative Policy Optimization (GRPO).** GRPO (Shao et al., 2024) is an online Policy Gradient method that discards the critic model to save computation. To estimate the advantage, it samples a group of outputs $(\mathbf{y}_1, \ldots, \mathbf{y}_G)$ from the old policy $\pi_{\theta_{\text{old}}}$ for each input $\mathbf{x}$. The advantage for the $g$-th sample in a group is computed by normalizing its reward against the group's reward distribution $\{r_1, \ldots, r_G\}$:

$$A_g = \frac{r_g - \text{mean}(\{r_1, \ldots, r_G\})}{\text{std}(\{r_1, \ldots, r_G\})}. \tag{5}$$

We consider verifiable reward functions in this paper. The policy is then optimized with the clipped objective:

$$\mathcal{L}_{\text{GRPO}}(\theta) = -\frac{1}{G} \sum_{i=1}^{G} \left[ \min \left( \frac{\pi_\theta(\mathbf{y}_i|\mathbf{x})}{\pi_{\theta_{\text{old}}}(\mathbf{y}_i|\mathbf{x})} A_i, \text{clip}\left( \frac{\pi_\theta(\mathbf{y}_i|\mathbf{x})}{\pi_{\theta_{\text{old}}}(\mathbf{y}_i|\mathbf{x})}, 1-\epsilon, 1+\epsilon \right) A_i \right) - \beta D_{\text{KL}}(\pi_\theta \| \pi_{\text{ref}}) \right]. \tag{6}$$

### 3.2 CONDITIONAL DECISION ENTROPY

The reasoning quality is difficult to optimize in hateful meme detection, as there is no reliable reward model due to the scarce rationale corpora and subjective human judgements. To address this, we propose Conditional Decision Entropy (CDE) as a proxy measure. The principle of CDE is straightforward: good reasoning should lead to a sharp and correct decision, while poor reasoning produces confusion.

**CDE Definition.** For an input $\mathbf{x}$, the LMM $\pi_\theta$ generates an explanation and decision response $\mathbf{y} = (\mathbf{e}, d) \sim \pi_\theta(\cdot \mid \mathbf{x})$ in the format of Eq. 2, where the final decision is sampled conditioned on the explanation and input $d \sim \pi_\theta(\cdot \mid \mathbf{e}, \mathbf{x})$. We define CDE as the entropy of the decision *conditioned on* the produced explanation:

$$H(d \mid \mathbf{e}, \mathbf{x}) = -\mathbb{E}_{d \sim \pi_\theta(\cdot|\mathbf{e},\mathbf{x})} \left[ \log \pi_\theta(d \mid \mathbf{e}, \mathbf{x}) \right]. \tag{7}$$

**Monte Carlo Estimator for CDE** To evaluate reasoning quality with CDE, we estimate the average CDE over the validation set. For each example $\mathbf{x}_i$, we sample $K = 16$ explanations $\mathbf{e}_{ik}$ with the policy $\pi_\theta$ and compute the entropy of the decision distribution. The estimator is

$$\widehat{H}(d \mid \mathbf{e}, \mathbf{x}) = \frac{1}{K|\mathcal{D}|} \sum_{i=1}^{|\mathcal{D}|} \sum_{k=1}^{K} H(d \mid \mathbf{e}_{ik}, \mathbf{x}_i), \quad \mathbf{e}_{ik} \sim \pi_\theta(\cdot \mid \mathbf{x}_i). \tag{8}$$

In the binary classification case, we experimented with collapsing the decision vocabulary to $\mathcal{V} \in \{\text{Yes}, \text{No}\}$, making CDE equivalent to binary entropy. We observed no significant difference compared to using the full vocabulary. For generalizability to fine-grained multi-class labels, we therefore adopt the full vocabulary formulation. A full derivation is provided in Appendix G.

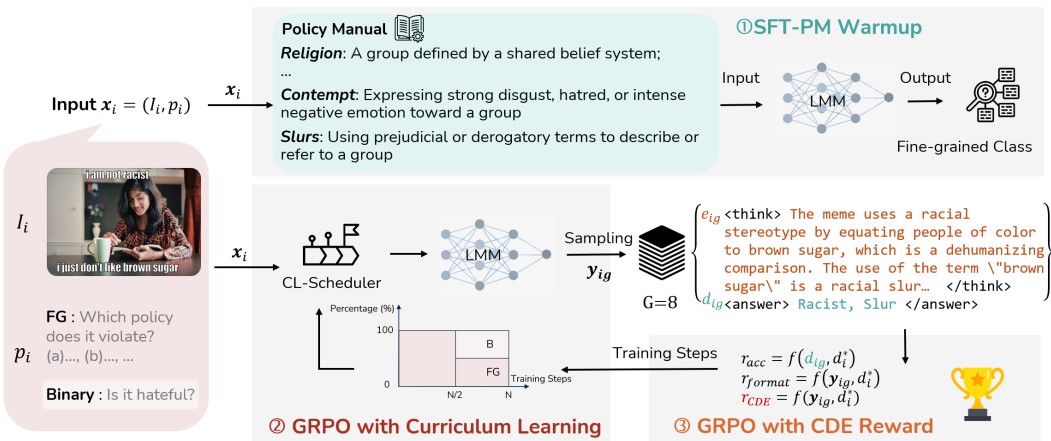

Figure 2: Architecture of ExPO-HM. Our framework consists of three key components: ① **SFT-PM Warmup.** The VLM is first trained with SFT using structured policy manuals derived from fine-grained labels and dataset guidelines, teaching the model to align decisions with explicit moderation policies. ② **GRPO with Curriculum Learning**. Training follows a two-stage schedule: the first 50% of steps use fine-grained data only for reasoning exploration, and the remaining 50% use a balanced 50/50 mix of fine-grained and binary data. ③ **GRPO with CDE reward**. In addition to the format reward ($r_{\text{format}}$) and accuracy ($r_{\text{acc}}$) reward used in standard GRPO, we also add a Conditional Decision Entropy ($r_{\text{CDE}}$) reward.

## 3.3 ExPO-HM Framework

Inspired by human moderator training, where annotators first study annotation guidelines and then practice applying them to tasks of increasing difficulty, ExPO-HM, as shown in Figure 2, first learn policy knowledge through SFT, then refines its reasoning via GRPO with curriculum learning, progressing from fine-grained to binary classification.

**SFT Warmup on Structured Policy Manuals (SFT-PM).** We first teach the LMM moderation policy knowledge by converting each dataset's fine-grained labels into a structured policy manual as the input prompt. Descriptions derived from the dataset annotation guidelines are added to each policy item in the policy manual. Details of this conversion process are provided in Appendix B.2. We optimize the language modelling loss in Eq. 3 with this policy manual augmented input for each meme, and the target response $\mathbf{y}^*$ is the fine-grained label $d_i^*$. Note that we do not use human-written gold hateful explanation $e^*$ in the warmup stage, as they are off-policy and lead to worse performance, which we discuss in Sec 4.5.

**GRPO with Curriculum Learning (GRPO-CL).** After the SFT-PM warmup, we conduct GRPO curriculum learning. We begin with fine-grained classification to incentivize policy understanding through diverse reasoning exploration, then introduce binary classification for hateful vs. benign detection. We test various curriculum schedulers, switching after fine-grained accuracy plateaus, adjusting the budget split between stages, or adjusting the mixing ratio of the fine-grained and binary data in the second stage, and find similar performance as long as fine-grained reasoning precedes binary. We therefore adopt a simple 50/50/50 strategy: the first 50% of steps use fine-grained data only, and the remaining 50% use a balanced 50/50 mix of fine-grained and binary data.

We optimize the clipped surrogate loss in Eq. 6 using the group-relative advantage in Eq. 5. The reward $r_{ig}$ corresponds to the $g$-th response in the sampled group for the $i$-th training example

$$r(\mathbf{y}_{ig}, d_i^*) = r_{\text{format}} + r_{\text{acc}} + w\, r_{\text{CDE}}, \tag{9}$$

where $r_{\text{format}} \in \{0, 1\}$ checks if the output obeys the correct template in Eq. 2. The accuracy reward $r_{\text{acc}} \in [0, 1]$ measures prediction correctness with partial credit for multi-class fine-grained classification and penalties for over-prediction. For binary classification, it requires an exact match and thus $r_{\text{acc}} \in \{0, 1\}$. For the GRPO baseline, we set $w = 0$, leaving only the format and accuracy rewards. Now let's define CDE Reward $r_{\text{CDE}}$.

**CDE as a Reward** Although GRPO with curriculum learning improves over the naive GRPO baseline, it still falls short in producing reliable reasoning. As introduced in Sec. 3.2, CDE provides a proxy for reasoning quality. If the prediction is sharp and correct, the reasoning is helpful and should be rewarded; if it is wrong but confident, the reasoning is misleading and should be penalized. We therefore incorporate it as an additional reward to guide ExPO-HM.

For each group-sampled example $\mathbf{y}_{ig}$ of each input $\mathbf{x}_i$, we denote the CDE as $h_{ig}$ and correctness as $\delta_{ig}$:

$$h_{ig} = H(d \mid \mathbf{e}_{ig}, \mathbf{x}_i), \qquad \delta_{ig} = \mathbf{1}\big[d_{ig} = d_i^*\big]. \tag{10}$$

We reward confident correctness, tolerate uncertainty when wrong, and penalize confident errors. The CDE reward for the example $\mathbf{y_{ig}}$ is

$$r_{\mathrm{CDE}}(h_{ig}, \delta_{ig}) = \delta_{ig} \cdot \begin{cases} w, & h \leq a \\ w\dfrac{b - h_{ig}}{b - a}, & a < h_{ig} < b \\ 0, & h_{ig} \geq b \end{cases} + (1 - \delta_{ig}) \cdot \begin{cases} -\rho w, & h_{ig} \leq a \\ w\dfrac{h_{ig} - a}{b - a}, & a < h_{ig} < b \\ w, & h_{ig} \geq b \end{cases} \tag{11}$$

CDE rewards contribute a maximum of weight $w$, with $\rho$ controlling the penalty strength for over-confident wrong predictions. Unless otherwise noted, we use default hyperparameters $a = 0.1$, $b = 0.5$, $w = 0.2$, and $\rho = 0.25$. A detailed hyperparameter analysis is provided in Appendix C.3. The $r_{\mathrm{CDE}}$ can thus be fed into Eq. 9 to obtain the reward to compute advantage to optimize the GRPO objective.

## 4 EXPERIMENTS

### 4.1 EXPERIMENTAL SETUP

**Dataset.** We evaluate the binary and fine-grained classification on three meme classification datasets: HatefulMemes (Kiela et al., 2020), MAMI (Fersini et al., 2022), and PrideMM (Shah et al., 2024).

**Tasks** We evaluate binary classification (hateful vs benign) on all three datasets. For fine-grained classification, we assess attack methods and target groups on HatefulMemes, attack methods on MAMI, and stance towards LGBTQ+, along with target group detection on PrideMM. Due to the scarcity of annotated hate rationales, we only evaluate reasoning quality on HatefulMemes, where gold human rationales are available (Hee et al., 2023). Detailed dataset descriptions and statistics are provided in Appendix B.

**Evaluation Metrics.** We evaluate classification tasks using macro F1 following prior work (Shah et al., 2024). For fine-grained classification, we use micro F1 due to a highly imbalanced class distribution. For reasoning quality, we adopt the LLM-as-a-judge method (Yang et al., 2023; Mei et al., 2025) to measure alignment between model-generated and human rationales. The detailed evaluation setup is provided in Appendix D. We further include different LLM judge experiments in Appendix E. In addition, we report CDE as a proxy to reasoning quality and verify its correlation with LLM-as-a-judge in Section 4.3. We also report human evaluation of the reasoning in Section 4.7

### 4.2 BASELINES

We compare ExPO-HM with comprehensive baselines on Qwen2.5-VL-3B and Qwen2.5-VL-7B (Bai et al., 2025) in Table 1. In this section, we describe the baseline setup briefly. Full implementation details are provided in Appendix C to ensure reproducibility.

**SFT.** In this paper, we consider two variants of SFT as baselines. *Direct-SFT* is trained with the ground-truth label as the target ($\mathbf{y}^* = d^*$ in Eq. 3), while *CoT-SFT* uses Explain-then-Detect prompt adopted in DPO and GRPO, where the target sequence is the chosen response in DPO sampling ($\mathbf{y}^* = \mathbf{y}^+$ in Eq. 3). In practice, we find that *Direct-SFT* consistently outperforms *CoT-SFT*, even when inference is performed with the Explain-then-Detect prompt. We therefore report *Direct-SFT* as the default baseline. For classification, we train and report separate models based on the binary

and the fine-grained subset, and report the best results. Full results for each model are provided in Table 3, while Table 1 reports the best system.

**DPO & GRPO.** For DPO and GRPO, we initialize from the fine-grained SFT warmup, but without the policy-manual style augmentation. We sweep different $\beta$ values in DPO to get the best performance on the validation set. The GRPO baseline is trained with the same compute budget as ExPO-HM, using identical hyperparameter settings in both the warmup and GRPO fine-tuning stages.

**Best prior systems.** We compare ExPO-HM with the best prior systems. RA-HMD (Mei et al., 2025) is the state-of-the-art direct detection model, combining two-stage fine-tuning and retrieval-augmented classification. Although primarily designed for direct detection, it supports reasoning evaluation via prompting, so we report its LLM-as-a-judge scores. All RA-HMD results are based on Qwen2.5-VL-7B (Bai et al., 2025). For Explain-then-Detect, we compare two recent systems: LOREHM (Huang et al., 2024), a reflective reasoning agent with tool-calling capability built on LLaVA-Next-34B (Liu et al., 2024), and U-CoT+ (Pan et al., 2025), which uses human-guided CoT prompting with Qwen2-VL-7B (Wang et al., 2024a) for meme-to-text conversion and Qwen2.5-14B (Qwen et al., 2025) for answer generation. We can only report their results on the binary classification due to their prompt-based agent design, these systems cannot be directly adapted for fine-grained classification or structured reasoning tasks. Furthermore, we did not include closed-source reasoning LMMs such as the OpenAI o-series (OpenAI, 2024) as baselines, since over 30% of requests were blocked by the API server due to the harmful nature of the examples.

Table 1: Comparing ExPO-HM with baseline systems across three datasets. *B* stands for Binary and *R* stands for Reasoning. LLM refers to the LLM-as-a-judge score. Best results are in **bold**. ↑ indicates higher is better, ↓ lower is better.

| # Model | **HatefulMemes** | | | | | **MAMI** | | | **PrideMM** | | | |
|---|---|---|---|---|---|---|---|---|---|---|---|---|
| | *B* | *Attack* | *Target* | *R* | | *B* | *Attack* | *R* | *B* | *Stance* | *Target* | *R* |
| | F1 ↑ | F1 ↑ | F1 ↑ | LLM ↑ | CDE ↓ | F1 ↑ | F1 ↑ | CDE ↓ | F1 ↑ | F1 ↑ | F1 ↑ | CDE ↓ |
| *Direct Detection Baselines* | | | | | | | | | | | | |
| 1 Qwen2.5-VL-3B | | | | | | | | | | | | |
| 2   *Zero-shot* | 53.1 | 42.1 | 60.1 | - | - | 61.1 | 48.2 | - | 58.6 | 53.7 | 48.8 | - |
| 3   *SFT* | 71.9 | 64.3 | 69.3 | - | - | 77.9 | 61.8 | - | 74.3 | 58.6 | 53.2 | - |
| 4 Qwen2.5-VL-7B | | | | | | | | | | | | |
| 5   *Zero-shot* | 59.8 | 50.3 | 60.2 | - | - | 63.4 | 50.2 | - | 65.2 | 56.8 | 51.1 | - |
| 6   *SFT* | 75.0 | 64.7 | 71.1 | - | - | 78.1 | 63.1 | - | 75.6 | 60.2 | 61.0 | - |
| 7   *RA-HMD* | 80.2 | - | - | 5.5 | - | 81.0 | - | - | 77.8 | - | - | - |
| *Explain-then-Detect Systems* | | | | | | | | | | | | |
| 8 LOREHM (34B) | 65.6 | - | - | - | - | 75.3 | - | - | - | - | - | - |
| 9 U-CoT+ (14B) | 72.4 | - | - | - | - | 79.9 | - | - | 71.4 | - | - | - |
| 10 Qwen2.5-VL-3B | | | | | | | | | | | | |
| 11   *Zero-shot* | 52.5 | 41.7 | 58.7 | 3.3 | 0.42 | 58.7 | 41.7 | 0.32 | 52.6 | 51.2 | 40.8 | 0.33 |
| 12   *SFT* | 62.3 | 62.7 | 63.3 | 3.6 | 0.40 | 69.2 | 60.1 | 0.34 | 63.2 | 56.6 | 49.8 | 0.29 |
| 13   *DPO* | 59.6 | 52.3 | 58.1 | 3.5 | 0.42 | 66.8 | 50.2 | 0.36 | 64.2 | 55.5 | 48.9 | 0.34 |
| 14   *GRPO* | 63.4 | 55.6 | 66.1 | 3.8 | 0.32 | 76.6 | 61.2 | 0.19 | 72.1 | 57.3 | 48.4 | 0.18 |
| 15   *ExPO-HM* | 74.7 | 71.5 | 73.7 | 5.1 | 0.16 | 80.7 | 70.4 | 0.08 | 75.6 | 66.5 | 62.1 | 0.12 |
| 16 Qwen2.5-VL-7B | | | | | | | | | | | | |
| 17   *Zero-shot* | 65.9 | 44.7 | 64.5 | 5.0 | 0.33 | 63.9 | 46.5 | 0.23 | 59.4 | 54.6 | 50.2 | 0.28 |
| 18   *SFT* | 74.5 | 58.4 | 69.4 | 5.0 | 0.33 | 72.8 | 62.6 | 0.19 | 68.3 | 58.0 | 50.9 | 0.28 |
| 19   *DPO* | 73.6 | 63.2 | 66.6 | 4.9 | 0.32 | 72.3 | 56.6 | 0.22 | 69.5 | 56.3 | 52.3 | 0.30 |
| 20   *GRPO* | 74.5 | 61.2 | 64.5 | 5.2 | 0.26 | 76.8 | 63.7 | 0.09 | 73.2 | 58.6 | 60.1 | 0.14 |
| 21   *ExPO-HM* | **81.1** | **75.6** | **77.2** | **6.2** | **0.03** | **82.3** | **73.0** | **0.04** | **78.7** | **68.4** | **65.1** | **0.08** |

## 4.3 COMPARING EXPO-HM TO BASELINE SYSTEMS

Table 1 compares ExPO-HM with the aforementioned baseline post-training methods and state-of-the-art systems. We report qualitative examples and error cases in Appendix J. Here, we summarize the key observations.

**Baseline Explain-then-Detect methods hurt classification performance.** Under *Explain-then-Detect*, post-training variants (SFT/DPO/GRPO, #18-#20 for Qwen2.5-VL-7B) consistently underperform the Direct-Detection SFT baseline (#6), except for the comparable performance on the MAMI Attack classification. Larger agentic and CoT systems (LOREHM, U-CoT+, #8-#9) also fall short of strong Direct-Detection baselines like SFT and RA-HMD (#6-#7 ). For instance, the binary classification on HatefulMemes is 80.2 on RA-HMD vs 72.4 F1 with U-CoT+. On HatefulMemes binary classification, RA-HMD reaches 80.2 F1, compared to 72.4 with U-CoT+. Explain-then-Detect systems are crucial for building automatic moderation systems that can truly support real-world moderators, but these results highlight that simply adding explicit rationales through CoT prompting or standard post-training hurts classification accuracy. This motivates the design of ExPO-HM, which aims to improve Explain-then-Detect systems without sacrificing predictive performance.

**Naive post-training barely improves performance.** Explain-then-Detect post-training (#18–#20) improves classification over zero-shot (#17), but reasoning quality stagnates, failing to meet the goal of improving reasoning through post-training. On HatefulMemes with Qwen2.5-VL-7B, the zero-shot LLM-as-a-judge score is 5.0; DPO drops below this, while GRPO only nudges it to 5.2. Even with online RL, reasoning remains difficult to improve. Moreover, post-training still underperforms strong CoT systems specifically designed for hateful meme detection (#8-#9). This underscores the need for dedicated post-training methods like ExPO-HM, which are tailored to hateful meme detection and designed to improve not only classification accuracy but also the quality of explanations.

**ExPO-HM consistently outperforms.** ExPO-HM delivers the strongest performance across binary detection, fine-grained classification, and reasoning. On Qwen2.5-VL-7B, it surpasses RA-HMD and all post-training baselines, achieving large gains in fine-grained F1 (**+14.4** on HatefulMemes Attack, **+12.7** on Target, compared to GRPO with equal compute). Reasoning also improves markedly, with 6.2 on LLM-as-a-judge vs. 5.2 for GRPO. In Appendix E, we conduct additional evaluations using different LLM judges and paraphrased prompts, and ExPO-HM consistently outperforms all baselines under all settings. Appendix I further reports per-class metrics for Attack and Target detection on HatefulMemes, showing that ExPO-HM achieves consistent improvements, particularly on the most challenging categories. These results confirm ExPO-HM's effectiveness across datasets and tasks.

**Strong correlation between LLM-as-a-judge metric and CDE metric.** On the HatefulMemes reasoning dataset, we observe a strong alignment between the LLM-as-a-judge score and the CDE score. To quantify this, we evaluate the correlation based on results from all the reported setups, with three random seeds each, yielding 60 data points. We find a strong negative correlation (Pearson $r = -0.78$, Spearman $\rho = -0.81$, both $p < 0.001$), confirming that lower CDE values, reflecting more confident and accurate reasoning, correspond to higher reasoning quality.

### 4.4    ABLATION STUDY OF EXPO-HM COMPONENTS

We conduct an ablation study to examine the contribution of the three key components in ExPO-HM. Results on HatefulMemes with Qwen2.5-VL-7B are reported in Table 2. Without SFT-PM, the warmup falls back to SFT with fine-grained labels without policy manual augmentation. Without GRPO-CL, GRPO is trained on a randomly

Table 2: Ablation Study of ExPO-HM Components.

| | **Components** | | | **HatefulMemes** | | | | |
|---|---|---|---|---|---|---|---|---|
| # | SFT-PM | GRPO-CL | CDE | *B* F1 ↑ | *Attack* F1 ↑ | *Target* F1 ↑ | *R* LLM ↑ | CDE ↓ |
| 1 | - | - | - | 74.5 | 61.2 | 64.5 | 5.2 | 0.263 |
| 2 | ✓ | - | - | 75.8 | 70.8 | 70.2 | 5.6 | 0.092 |
| 3 | ✓ | ✓ | - | 78.4 | 74.3 | 76.1 | 5.8 | 0.056 |
| 4 | ✓ | ✓ | ✓ | 81.1 | 75.6 | 77.2 | 6.2 | 0.026 |

mixed set of binary and fine-grained data. Without CDE, GRPO uses only the format and accuracy rewards.

**SFT-PM enhanced the fine-grained warmup.** Compared to the baseline warmup without policy manual augmentation (#1), SFT-PM improves performance across all metrics. This indicates that

Table 3: Comparing SFT warmup variants on HatefulMemes on Qwen2.5-VL-7B: no warmup (-), SFT on binary labels (SFT-B), SFT on gold reasoning (SFT-R), SFT on fine-grained labels (SFT-FG), and SFT with policy-manual augmentation (SFT-PM).

| # | Warmup | SFT | | | | w/ GRPO-CL and CDE | | | |
|---|---|---|---|---|---|---|---|---|---|
| | | *B* F1 ↑ | *Attack* F1 ↑ | *Target* F1 ↑ | *R* LLM ↑ | *B* F1 ↑ | *Attack* F1 ↑ | *Target* F1 ↑ | *R* LLM ↑ |
| 1 | - | 65.9 | 44.7 | 64.5 | 5.0 | 73.3 | 69.3 | 72.1 | 5.2 |
| 2 | SFT-B | 74.1 | 58.2 | 69.4 | 4.9 | 73.5 | 66.8 | 70.1 | 5.1 |
| 3 | SFT-R | 72.2 | 51.6 | 63.1 | 5.0 | 79.2 | 72.3 | 73.2 | 5.7 |
| 4 | SFT-FG | 72.5 | 58.4 | 67.7 | 4.9 | 78.9 | 73.4 | 73.4 | 5.6 |
| 5 | SFT-PM | 74.3 | 64.6 | 68.8 | 5.0 | 81.1 | 75.6 | 77.2 | 6.2 |

fine-grained labels alone are insufficient for policy understanding, while policy manual augmentation substantially strengthens both classification and reasoning. In Sec. 4.5, we further present a systematic comparison of different warmup strategies.

**GRPO-CL further improves performance.** Building on SFT-PM, adding curriculum learning to GRPO (#3) yields further gains across the board. The key difference is ordering, GRPO-CL first lets the model explore reasoning over the fine-grained labels before binary classification. This order proves crucial: standard GRPO produces short average responses (28 tokens) in binary classification, while GRPO-CL nearly doubles this (52 tokens), indicating not only higher quality but also more detailed reasoning is incentivized during training.

**CDE improves both accuracy and explanation quality.** Adding CDE on top of SFT-PM and GRPO-CL further improves the performance. Notably, LLM-judge score improved to 6.2, and a marked drop in CDE 0.026, suggesting that the model's rationales become more aligned with sharp, correct decisions.

## 4.5 EFFECTS OF DIFFERENT WARMUP STRATEGY

Table 3 compares five warmup strategies for Qwen2.5-VL-7B on the HatefulMemes dataset. For each, we report the Explain-then-Detect performance after SFT and the performance after GRPO-CL with CDE reward.

**Good SFT does not necessarily transfer to good RL performance.** Although SFT-B performs better than SFT-R and SFT-FG at the SFT stage, its performance after GRPO-CL is comparably worse than its counterparts, even below the no-warmup baseline. This suggests that binary-only warmup fails to equip the model with the moderation concepts needed for reasoning-guided RL. In contrast, our proposed SFT-PM explicitly teaches such concepts via policy manual augmentation, yielding both stronger SFT performance and the best results after ExPO-HM training.

## 4.6 CDE ANALYSIS

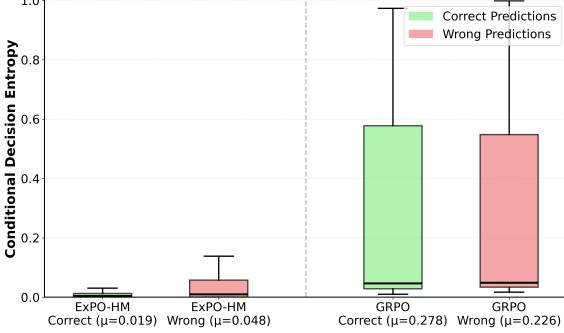

Figure 3: Comparison of CDE distributions between ExPO-HM and GRPO on the HatefulMemes validation set with Qwen2.5-VL-7B.

**CDE Distribution Analysis.** Figure 3 presents box-and-whisker plots of CDE distributions for ExPO-HM and GRPO on the HatefulMemes validation set with Qwen2.5-VL-7B. ExPO-HM maintains very low CDE for correct predictions ($\mu = 0.019$) and higher CDE for wrong ones ($\mu = 0.048$), yielding a clear separation. In contrast, the GRPO baseline shows high CDE for both correct ($\mu = 0.278$) and wrong predictions ($\mu = 0.226$), showing weaker separation. This demonstrates that ExPO-HM produces reasoning that is not only more accurate but also better aligned with decision confidence.

**CDE for Decision Calibration.** The CDE reward penalizes confident wrong predictions and rewards confident correct decisions. This encourages a well-behaved decision distribution: the model becomes confident only when it is likely to be correct, and becomes uncertain when the outcome is ambiguous.

To validate whether CDE reward improves calibration, we compute Expected Calibration Error (ECE) and Brier score using the model's probability assigned to the final answer token, conditioned on the generated explanation under the Explain-then-Detect setup. We observe that ExPO-HM consistently improves calibration compared to the GRPO baseline. Notably, for Qwen2.5-VL-3B, ExPO-HM reduces the Brier score from 0.590 to 0.283, indicating substantially more reliable decision confidence. Full calibration results are reported in Appendix H.1. We additionally provide a derivation of the upper bound on the Brier score under the ideal ExPO-HM policy in Appendix H.2.

**CDE and Policy Entropy.** We test whether adding the CDE reward causes policy entropy collapse, a phenomenon reported in prior RL work (Cui et al., 2025) when entropy bonuses are removed. Our results show that overall policy entropy remains comparable to the baseline GRPO system without CDE, confirming that the CDE reward, acting only on the decision part of the generation, does not reduce exploration.

**CDE Hyperparameters.** For CDE hyperparameters, we conduct standard hyperparameter tuning via grid search on the HatefulMemes validation set. Once the optimal values were identified, we fixed these parameters and applied them directly to MAMI and PrideMM. We observe that as long as the hyperparameters fall within a reasonable range, the model performance remains highly stable. We provide the detailed insights of hyperparameter tuning in Appendix C.3.

### 4.7 HUMAN EVALUATION OF MODEL-GENERATED REASONING

To assess the quality of the generated reasoning beyond LLM-as-a-judge evaluation, we further conduct two complementary human evaluations. Each example is independently evaluated by three crowd-sourced annotators with at least an undergraduate degree and demonstrated familiarity with internet meme culture. We evaluate both the GRPO baseline and our ExPO-HM model. The detailed evaluation setup and full results are provided in Appendix F.

**Coherence Evaluation.** Annotators judge whether the model's final decision is logically supported by its rationale. The GRPO baseline achieves 96% coherent outputs, whereas ExPO-HM attains 100% coherence.

**Helpfulness Evaluation.** Annotators also rate how helpful each rationale is for understanding why the meme is hateful or benign, using a 0–4 Likert scale following prior work (Wang et al., 2024b). We obtain average helpfulness scores of 1.6 for GRPO and 2.2 for ExPO-HM. After normalizing the scores to the same 0–10 scale used by the LLM-as-a-judge (4.1 vs. 5.5), we observe high agreement between human and LLM evaluations in the relative improvement from GRPO to ExPO-HM.

## 5 CONCLUSION

We propose ExPO-HM, which combines SFT warmup on policy-manual–augmented data with GRPO curriculum learning, guided by a Conditional Decision Entropy reward to promote high-quality reasoning. Comprehensive experiments show that ExPO-HM achieves state-of-the-art performance on binary detection, fine-grained classification, and reasoning quality.

## ETHICAL STATEMENT

**Societal benefits.** Hateful meme detection systems such as ExPO-HM can help automatically identify and mitigate harmful online content, reducing the prevalence of hate speech. By providing explanations in addition to predictions, our system not only supports safer digital environments for end-users but also alleviates the burden on human content moderators, improving their well-being. We believe such systems play an essential role in fostering respectful online communication and contributing to healthier digital communities.

**Intended use.** We will enforce strict access controls for releasing model checkpoints and artifacts. Access will be limited to researchers who agree to our terms of use, which explicitly restrict the system to the detection and prevention of hateful speech. Any use that promotes, condones, or encourages hate speech or other harmful content is strictly prohibited.

**Misuse potential.** Although ExPO-HM is not designed to introduce bias, it is trained on datasets that may reflect societal or annotator biases (Pramanick et al., 2021). These biases could propagate into model predictions. To mitigate risks of unfair or disproportionate moderation, human oversight remains essential when deploying such systems.

**Deployment considerations.** Moderation of hateful content is inherently influenced by cultural norms and subjective judgments. Expressions considered benign in one context may be offensive in another. Since ExPO-HM is trained with policy manuals, its outputs depend critically on the underlying moderation policies. Careful review and adaptation of community guidelines are crucial to ensure responsible deployment across diverse cultural and linguistic contexts.

**Usage of Datasets.** The datasets used in this study, HatefulMemes, MAMI, and PrideMM, were curated for research purposes to combat online hate speech. We strictly adhere to the terms of use established by the dataset authors.

## ACKNOWLEDGMENTS

Jingbiao Mei is supported by the Cambridge Commonwealth, European and International Trust for his PhD studies in Engineering at the University of Cambridge. This work was conducted during an internship at Xiaohongshu Inc.

Jinghong Chen is supported by the Warwick Postgraduate Studentship from Christ's College and the Huawei Hisilicon Studentship for the undertaking of the PhD in Engineering at the University of Cambridge.

Prof. Bill Byrne holds concurrent appointments as a Professor of Information Engineering at Cambridge University and as an Amazon Scholar. This publication describes work performed at Cambridge University and is not associated with Amazon.

We would also like to thank all the reviewers for their knowledgeable reviews.

## REPRODUCIBILITY STATEMENT

We provide detailed dataset usage and statistics in Appendix B, including all augmentation procedures introduced in this paper. For the evaluation of reasoning, we detail the process in Appendix D. Experimental setups, implementation details including code base usage, software version, hardware configuration, and hyperparameter settings are described in Appendix C to facilitate reproducibility. Code and data available: https://github.com/JingbiaoMei/ExPO-HM

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

## A    USE OF LARGE LANGUAGE MODELS

Our coding work was assisted by Github Copilot. OpenAI ChatGPT was only used for proofreading and spell-checking. We claim that the content presented in this paper was fully original.

## B    DATASET DETAILS AND STATISTICS

### B.1    DATASET DETAILS AND PROMPT USED

**HatefulMemes.**    Released by Meta in 2020, HatefulMemes (Kiela et al., 2020) contains 12,000 memes annotated as hateful or benign by trained experts. This benchmark dataset synthesizes memes targeting religion, race, disability, and gender. It includes confounder examples where the benign memes are generated by altering either the image or text to challenge models' ability in multimodal reasoning.

In 2021, Meta further released an extended annotation set, the Fine-Grained Hateful Meme Classification dataset (Mathias et al., 2021a). This dataset augments the original HatefulMemes with detailed labels for both attack types and target groups, enabling fine-grained classification.

For fine-grained classification training and evaluation, except for the SFT-PM training, we adopt the following prompts for *attack types*:

> Does this meme express hate or negative bias toward any of the following protected groups?
> Religion, Race, Sex, Nationality, Disability
> Please respond with one or more of the protected categories if applicable. If the meme does not contain any hateful content, respond with Benign.

For *target groups*:

> Does this meme use any of the following types of attack against a group?
> Dehumanizing, Inferiority, Inciting violence, Mocking, Contempt, Slurs, Exclusion
> Please respond with one or more of the attack types if applicable. If the meme does not contain any hateful content, respond with Benign.

**MAMI.**    The MAMI dataset (Fersini et al., 2022) focuses on detecting misogynistic memes sourced from various social media platforms, including Twitter and Reddit, as well as meme creation and sharing websites, and even anti-women websites and forums. It contains annotations for two tasks: (1) binary classification of misogyny and (2) categorization of misogyny types. In this work, we use the binary task to detect whether a meme is misogynistic and treat the type categorization as a fine-grained multi-class, multi-label classification problem. Each sample can take one or more of four attack type labels: objectification, shaming, stereotype, and violence, or Benign if no harm is present.

For all fine-grained training and evaluation tasks on fine-grained classes except for the SFT-PM training, we use the prompt:

> Does this meme use any of the following types of attack against a group?
> objectification, shaming, stereotype, violence
> Please respond with one or more of the attack types if applicable.
> If the meme does not contain any hateful content, respond with Benign.

**PrideMM.**    PrideMM (Shah et al., 2024) contains LGBTQ+ themed memes annotated for four tasks: hate speech detection, hate target identification, topical stance classification, and humor detection. In this work, we use the hate speech classification annotations for the binary hateful meme detection.

We further use the hate target identification and topical stance classification in our fine-grained classification setup. Both tasks are formulated as multi-class, single-label classification.

For all fine-grained training and evaluation tasks except for the SFT-PM training, we use the *Target* identification prompt:

> Based on the content and context of this meme, does this meme express hate or negative bias toward any of the following targets?
>
> Choose from the following options: undirected individual community organization
>
> If the meme does not contain any hateful content, respond with Benign.
> What is the target type?

*Stance* classification prompt:

> Based on the content and context of this meme, what is the stance towards LGBTQ+ individuals or communities?
>
> Choose from the following options: neutral, support, oppose
>
> What is the stance?

**Explain-Then-Detect Prompt.** The above shows the prompt used for the direct detection setup. For the Explain-Then-Detect Prompt, we adapt the prompt from the VeRL training library:

> `<BASE_PROMPT>` Output the thinking process in `<think> </think>` and final answer in `<answer> </answer>` tags. The output format should be as follows: `<think> ... </think> <answer>...</answer>` Please strictly follow the format.

The `<BASE_PROMPT>` is replaced with the specific prompts defined above.

## B.2 POLICY MANUAL CONSTRUCTION

For HatefulMemes, MAMI, and PrideMM, the dataset authors provide detailed annotation guidelines in prose, including lists of protected or offensive categories (e.g., ethnicity, race, violence) and their definitions. We extract this information and convert it into a concise bullet-point list, which we refer to as the policy manual. Below is an example of the original annotation guideline from (Kiela et al., 2020):

> *"A direct or indirect attack on people based on characteristics, including ethnicity, race, nationality... We define attack as violent or dehumanizing (comparing people to non-human things, e.g. animals) speech, statements of inferiority..."*

Representing the annotation guidelines in the structured Policy Manual form makes it easier to create targeted instruction-following SFT data compared to using long-form prose descriptions. The conversion from guideline to policy manual is a one-time process performed by a human expert. Below, we detail the specific policy manuals for each dataset.

**HatefulMemes.** For policy-manual SFT, we use the following prompts:

> Does this meme use any of the following types of attack against a group?
> - Dehumanizing: Presenting a group as subhuman, explicitly or implicitly
> - Inferiority: Claiming that a group is inferior, less worthy, or less important
> - Inciting violence: Calling for or encouraging harm—physical or otherwise—towards a group
> - Mocking: Belittling or making jokes about a group
> - Contempt: Expressing strong disgust, hatred, or intense negative emotion
> - Slurs: Using prejudicial or derogatory terms to describe a group
> - Exclusion: Advocating for removal, segregation, or marginalization of a group
> If the meme does not use any of these attack types, respond with Benign.

> Does this meme express hate or negative bias toward any of the following protected groups?
> - Religion: A group defined by shared belief systems
> - Race: A group defined by racialized physical characteristics
> - Sex: A group defined by sexual attributes or sexual identification
> - Nationality: A group defined by country or region of origin
> - Disability: A group defined by conditions leading to permanent dependencies
> If the meme does not target any protected group, respond with Benign.

**MAMI.** For policy-manual SFT, we use the following prompts:

> Based on the content and context of this meme, does it use any of the following types of attack against a group?
> Choose from the following options:
> - objectification: The content reduces individuals or groups to objects, ignoring their personhood or agency
> - shaming: The content ridicules, mocks, or publicly humiliates individuals or groups
> - stereotype: The content attributes oversimplified, generalized, or exaggerated traits to individuals or groups
> - violence: The content depicts or encourages physical harm, threats, or violent actions against individuals or groups
> If the meme does not contain any hateful content, respond with Benign.
> What is the attack type?

**PrideMM.** For policy-manual SFT, we use the following prompts:

> Based on the content and context of this meme, does this meme express hate or negative bias toward any of the following targets?
>
> Choose from the following options:
> - undirected: General targeting without specific individuals or groups
> - individual: Targeting specific individuals
> - community: Targeting LGBTQ+ communities or groups
> - organization: Targeting specific organizations or institutions
> If the meme does not contain any hateful content, respond with Benign.
>
> What is the target type?

> Based on the content and context of this meme, what is the stance towards LGBTQ+ individuals or communities?
>
> Choose from the following options:
> - neutral: The content does not express clear support or opposition
> - support: The content expresses positive attitudes or support
> - oppose: The content expresses negative attitudes or opposition
>
> What is the stance?

### B.3 DATASET STATISTICS

**Binary Classification statistics.** Table 4 shows the data split for our binary evaluation datasets. For HatefulMemes, we use the `dev_seen` split as the validation set, `test_seen` as the test set.

**Fine-grained Classification Statistics.** Table 5 reports the detailed distribution of fine-grained attributes in the HatefulMemes dataset, covering both attack types and protected categories. Note that we use the `dev_unseen` split for final evaluation.

Table 6 provides the fine-grained label distributions for the MAMI dataset, focusing on Sub-task B (Type of Misogyny).

Table 4: Statistical summary of binary classification datasets.

| Datasets | Train | | Test | |
|---|---|---|---|---|
| | #Benign | #Hate | #Benign | #Hate |
| HatefulMemes | 5450 | 3050 | 500 | 500 |
| MAMI | 4500 | 4500 | 500 | 500 |
| PrideMM | 2581 | 2482 | 260 | 247 |

Table 5: Statistics of fine-grained attributes in the HatefulMemes dataset, showing attack types and protected categories across train, dev_unseen, and dev_seen splits.

| Fine-grained types | | train | dev_unseen | dev_seen |
|---|---|---|---|---|
| *Attack type* | dehumanizing | 1318 | 104 | 121 |
| | inferiority | 658 | 35 | 49 |
| | inciting_violence | 407 | 23 | 26 |
| | mocking | 378 | 29 | 35 |
| | contempt | 235 | 6 | 10 |
| | slurs | 205 | 4 | 6 |
| | exclusion | 114 | 8 | 13 |
| *Protected category* | religion | 1078 | 77 | 95 |
| | race | 1008 | 63 | 78 |
| | sex | 746 | 46 | 56 |
| | nationality | 325 | 20 | 26 |
| | disability | 255 | 17 | 22 |

Table 7 summarizes the fine-grained label distributions for the PrideMM dataset, including both target categories and stance annotations across the training and test splits.

**Hateful Reasoning Corpus Statistics.** Table 8 presents the dataset statistics for Hatred, which only includes hateful memes paired with explanations. The test set corresponds to the original `HatefulMemes dev_seen` split.

### B.4 DATASET LICENSES

To access the Facebook HatefulMemes dataset, one must follow the license from Facebook[1]. HarMeme and Harm-P are distributed for research purposes only, without a license for commercial use. MAMI is under Apache License 2.0. There is no specified license for PrideMM.

## C EXPERIMENT SETUP AND IMPLEMENTATION DETAILS

**Software Environment.** `PyTorch 2.5.1`, `CUDA 12.4`, Huggingface `Transformer 4.45.0` and `Python 3.10.12` were used for implementing the experiments. All the reported metrics were computed by `TorchMetrics 1.0.1`.

**Hardware Environment.** We conducted our GRPO and ExPO-HM experiments on a server equipped with 8 Nvidia H100 with 80GB of VRAM. For the DPO and SFT baselines, we use 1 GPU.

**Training Details** We freeze the vision module throughout fine-tuning, following the standard LMM fine-tuning protocol. We conduct all training with LoRA (Hu et al., 2022), with LoRA rank=64, $\alpha = 128$. For DPO sampling and all the inference, we use vLLM inference engine `0.9.2`.

---

[1] https://hatefulmemeschallenge.com/#download

Table 6: Statistics of Sub-task B in the MAMI dataset: type of misogyny labels across training and test sets. We treat this as a multilabel, multiclass fine-grained classification task.

| Category | Training Set | Test Set |
|---|---|---|
| Shaming | 1274 (25.48%) | 146 (29.20%) |
| Stereotype | 2810 (56.20%) | 350 (70.00%) |
| Objectification | 2202 (44.04%) | 348 (69.60%) |
| Violence | 953 (19.06%) | 153 (30.60%) |

Table 7: Statistics of fine-grained attributes in the PrideMM dataset, showing target categories and stance labels across training and test sets.

| Fine-grained types | | Train | Test |
|---|---|---|---|
| *Target* | Benign | 2208 | 260 |
| | Undirected | 666 | 68 |
| | Individual | 219 | 19 |
| | Community | 986 | 122 |
| | Organization | 249 | 38 |
| *Stance* | Neutral | 1252 | 140 |
| | Support | 1645 | 182 |
| | Oppose | 1431 | 185 |

## C.1  SFT AND DPO TRAINING

For Qwen2.5-VL fine-tuning, we employ the officially recommended fine-tuning library `LLaMA-Factory 0.9.3`[2] with official hyperparameter settings for all training tasks in both the SFT and DPO, except for the LoRA config that we mentioned above. For DPO, we sweep $\beta = 0.1, 0.3, 0.5, 0.7, 0.9$ and report the best results. For all runs, we train for 3 epochs, and then select the best checkpoint based on validation performance.

## C.2  GRPO TRAINING

We use the VeRL library `verl 0.4.1`[3]. We use the default hyperparameter settings for all training except for the LoRA configuration. For all runs, we train for 3 epochs, and then select the best checkpoint based on validation performance. The run time for ExPO-HM is about 4 hours on 8 GPUs, which is the same for the baseline GRPO experiment.

## C.3  CDE REWARD HYPERPARAMETER

For CDE hyperparameters, we conduct standard hyperparameter tuning via grid search on the HatefulMemes validation set. Once the optimal values were identified, we fixed these parameters and applied them directly to MAMI and PrideMM.

We observe that as long as the hyperparameters fall within a reasonable range, the model performance remains highly stable. Based on our hyperparameter tuning, we recommend the following default hyperparameters for new datasets: $a = 0.1$, $b = 0.5$, $w = 0.2$, and $\rho = 0.25$. The stable ranges, within which performance differences remain small, are: $0.05 \leq a \leq 0.15$, $0.4 \leq b \leq 0.6$, $0.15 \leq w \leq 0.25$, and $0.1 \leq \rho \leq 0.5$.

The following observations summarize the sensitivity characteristics of each CDE hyperparameter:

The following intuitions summarize the observed sensitivity patterns:

- **Low-entropy cutoff** ($a$)**:** If $a$ is too small (e.g., $< 0.05$), the reward provides limited benefit and tends to encourage overconfident predictions, leading to larger KL divergence

---

[2]https://github.com/hiyouga/LLaMA-Factory
[3]https://github.com/volcengine/verl/releases

Table 8: Statistics of the Hatred dataset, which only includes hateful memes with explanations. The test set corresponds to the original `HatefulMemes dev_seen` split.

|  | train | test | total |
|---|---|---|---|
| #Hatred (hateful memes only) | 2,982 | 246 | 3,228 |

and reduced policy entropy. If $a$ is too large (e.g., $0.20$), the CDE reward becomes too easy to satisfy, resulting in diminished improvement over the default value. A default value of $a = 0.1$ consistently performs well.

- **High-entropy cutoff** ($b$)**:** A small value of $b$ (e.g., $0.25$) excessively narrows the confidence band, pushing predictions into the overconfident region, significantly increasing KL divergence, reducing policy entropy, and destabilizing training. Conversely, overly large values make the reward too permissive and degrade performance. The recommended default value of $0.5$ achieves a balance between stability and effectiveness.

- **CDE weight** ($w$)**:** Large weights cause the CDE reward to dominate over the accuracy reward, which degrades performance when $w > 0.5$ compared to the default value of $0.2$. Extremely small weights, on the other hand, yield only marginal improvements over the baseline. A moderate value ($w = 0.2$) is most effective.

- **Penalty rate** ($\rho$)**:** Hyperparameter tuning for $\rho$ shows relatively low sensitivity. The default value works reliably across datasets. Performance degrades compared to the default setting when the penalty is removed (values near zero) or when it becomes overly strong (e.g., $\rho = 1.0$).

## D    EVALUATION OF MODEL-GENERATED REASONING

Following prior work (Yang et al., 2023; Mei et al., 2025), we assess explanation quality using an LLM judge. Specifically, we provide GPT-4o mini (`gpt-4o-mini-2024-07-18`) with reference explanations from (Hee et al., 2023). Following previous works (Mei et al., 2025), we adopt the same prompt:

> Compare the model-generated reasoning with the reference human reasoning for this hateful meme.
>
> Reference: {reference_reasoning}
> Model: {model_reasoning}
> Model Prediction: {model_prediction}
>
> Rate how well the model reasoning aligns with the reference on a scale of 0-10:
> - 9-10: Excellent alignment, captures all key points
> - 7-8: Good alignment, captures most key points
> - 5-6: Satisfactory alignment, captures some key points
> - 3-4: Poor alignment, misses many key points
> - 1-2: Very poor alignment, minimal understanding
> - 0: Completely wrong or unrelated
>
> Score: [0-10]
>
> Explanation: [1-2 sentences]

## E    ADDITIONAL RESULTS ON MODEL-GENERATED REASONING EVALUATION

In this section, we provide extended experiments and analysis for the LLM-as-a-judge evaluation used in our work. While LLM-based evaluators are widely adopted for assessing explanation qual-

ity and reasoning consistency, concerns remain regarding reproducibility, prompt sensitivity, and potential model deprecation. We address these concerns through (1) evaluation across multiple open-source judges, (2) inclusion of prompt-free metrics, and (3) sensitivity analysis with paraphrased prompts.

## E.1 EVALUATION ACROSS MULTIPLE LLM JUDGES

To ensure a fair comparison, we adopt the same judge model (`gpt-4o-mini-2024-07-18`) and the same evaluation prompt as prior work (Mei et al., 2025) in the main text. However, we acknowledge that evaluation with a commercial closed-source model may raise reproducibility concerns, particularly if the model is deprecated in the future. To address this, we conduct additional evaluations using open-source LLM judges with the same prompt for reproducibility, including `Qwen3-4B-Instruct-2507`, `gpt-oss-20b`, and `gemma-3-4b-it`. We further report scores using a newer and stronger closed-source judge GPT-5 (`gpt-5-2025-08-07`) for reference.

We also include BERTScore (Zhang* et al., 2020), which measures sequence-level semantic similarity between human rationales and model-generated explanations. Because BERTScore does not rely on prompts, it serves as a fully reproducible complement to LLM-based evaluations.

We compute BERTScore using the official implementation[4]. Following the same setup as our LLM-as-a-judge evaluation, we use the Hatred annotations from (Hee et al., 2023) as the reference rationales and compute the BERTScore for each model-generated explanation.

Table 9: Extended LLM-as-a-judge and BERTScore evaluation across models and training methods. Higher is better. The best-performing training method for each model is shown in **bold**.

| Model | GPT-4o mini | GPT-5 | Qwen3 | Gemma-3 | gpt-oss | BERTScore |
|---|---|---|---|---|---|---|
| **Qwen2.5-VL-3B** | | | | | | |
| Zero-shot | 3.3 | 1.5 | 2.0 | 2.9 | 2.4 | 0.52 |
| SFT | 3.6 | 2.0 | 2.4 | 3.8 | 2.6 | 0.53 |
| DPO | 3.5 | 2.2 | 2.3 | 3.7 | 3.0 | 0.53 |
| GRPO | 3.8 | 2.8 | 3.0 | 4.2 | 3.2 | 0.53 |
| ExPO-HM | **5.1** | **3.6** | **4.2** | **5.5** | **4.0** | **0.56** |
| **Qwen2.5-VL-7B** | | | | | | |
| Zero-shot | 5.0 | 3.8 | 4.5 | 5.2 | 4.0 | 0.55 |
| SFT | 5.0 | 4.0 | 4.6 | 5.5 | 3.9 | 0.55 |
| DPO | 4.9 | 3.5 | 4.5 | 4.9 | 3.6 | 0.54 |
| GRPO | 5.2 | 4.1 | 4.7 | 5.9 | 4.6 | 0.59 |
| ExPO-HM | **6.2** | **5.0** | **5.5** | **7.0** | **5.3** | **0.65** |

Table 9 shows the detailed results.

We note that different LLM judges exhibit different scoring distributions. BERTScore is less sensitive to subtle performance differences, likely due to its limited semantic understanding compared to modern LLMs. GPT-5, Qwen3, and gpt-oss tend to be more strict, while Gemma-3 tends to be more generous. Nevertheless, ExPO-HM models are consistently rated as the best-performing system under all evaluation metrics with substantial margins over GRPO models. This further validates the effectiveness of ExPO-HM.

## E.2 PROMPT SENSITIVITY ANALYSIS

To study prompt robustness, we manually paraphrased the evaluation prompt and re-evaluated all models using `gpt-4o-mini-2024-07-18`. As shown in Table 10, the results remain largely unchanged, suggesting that our evaluation is not sensitive to prompt phrasing.

Below is the paraphrased prompt for reference:

---

[4]`https://github.com/Tiiiger/bert_score`

Table 10: Prompt sensitivity analysis using GPT-4o mini as the judge. Higher is better.

| Model | Original Prompt | Paraphrased Prompt |
|---|---|---|
| **Qwen2.5-VL-3B** | | |
| Zero-shot | 3.3 | 3.2 |
| SFT | 3.6 | 3.4 |
| DPO | 3.5 | 3.3 |
| GRPO | 3.8 | 3.9 |
| ExPO-HM | **5.1** | **5.1** |
| **Qwen2.5-VL-7B** | | |
| Zero-shot | 5.0 | 5.0 |
| SFT | 5.0 | 5.2 |
| DPO | 4.9 | 5.0 |
| GRPO | 5.2 | 5.1 |
| ExPO-HM | **6.2** | **6.3** |

---

Evaluate how closely the model's explanation matches the human reference rationale for this hateful meme.

Reference: {reference_reasoning}
Model: {model_reasoning}
Model Prediction: {model_prediction}

Rate how well the model reasoning aligns with the reference on a scale of 0-10:
- 9-10: Outstanding alignment, captures all major points
- 7-8: Strong alignment, captures most important points
- 5-6: Moderate alignment, includes some relevant points
- 3-4: Weak alignment, overlooks many key points
- 1-2: Very weak alignment, shows little understanding
- 0: Completely incorrect or unrelated

If the model prediction is not hateful, which is incorrect, the highest score you may assign is 2.
Explanation: [1-2 sentences]
Score: [0-10]

---

## F   HUMAN EVALUATION ON MODEL-GENERATED REASONING

To assess the quality of the reasoning beyond LLM-as-a-judge, we further conduct two complementary human evaluations. Each example is independently evaluated by three crowd-sourced annotators with at least an undergraduate degree and demonstrated familiarity with internet meme culture. We evaluate both the GRPO baseline and our ExPO-HM model, and we randomize the order in which model outputs are presented to mitigate ordering or anchoring biases. An illustration of the annotation interface is provided in Figure 4.

### F.1   COHERENCE EVALUATION

Annotators judge whether the model's final decision is logically supported by its rationale. For each example, we ask:

> *"Does the model's final decision logically follow from its rationale? In other words, is the decision grounded in the explanation provided?"*

Annotators answer Yes or No. The GRPO baseline achieves 96% coherent outputs, while ExPO-HM achieves 100% coherence.

**Example 5**

Image ID: 06491

**Model A**

**Decision:** YES

**Rationale:**
The meme uses the term "tranny," which is a derogatory and offensive term used to refer to transgender people. This language is hurtful and dehumanizing.

**Model B**

**Decision:** YES

**Rationale:**
The meme contains the text "tranny for sale," which is a derogatory term used to refer to transgender individuals. This language is offensive and promotes harmful stereotypes and discrimination against the transgender community. The use of such language in a meme is generally considered hateful and disrespectful.

📋 **Ground Truth Annotation**

**Target Group(s): the lgbt community**
**Expert Reasoning:**
• dehumanizes the lgbt community as objects by suggesting that transgenders can be put on sale.

**Your Evaluation**

**Task 1: Coherence - Does the decision logically follow from the rationale?**

**Model A:**
○ Yes
○ No

**Model B:**
○ Yes
○ No

**Task 2: Helpfulness - How helpful is the rationale for understanding the meme?**

**Model A:**

| 0 Unrelated | 1 Weakly Related | 2 Partially Helpful | 3 Mostly Helpful | 4 Highly Helpful |
|---|---|---|---|---|

**Model B:**

| 0 Unrelated | 1 Weakly Related | 2 Partially Helpful | 3 Mostly Helpful | 4 Highly Helpful |
|---|---|---|---|---|

**Optional: Additional Comments**

Any additional observations or notes...

Figure 4: Human evaluation interface. Annotators assess the coherence and helpfulness of model-generated rationales, with access to the meme and ground-truth expert annotation.

### F.2 HELPFULNESS EVALUATION

Annotators also rate how helpful the rationale is for understanding why the meme is hateful. For each example, we ask the annotator:

> *"How helpful is the model's rationale for understanding why the meme is hateful or benign?"*

We adopt a 0–4 Likert scale (adapted from prior work (Wang et al., 2024b)), defined as:

- **0 — Unrelated / Incorrect:** Rationale is irrelevant, incorrect, or unusable; provides no moderation value.

- **1 — Weakly Related:** Touches on the content but lacks specificity or clarity; cannot support moderation.

- **2 — Partially Helpful:** Contains some relevant entities or violation cues but is incomplete or partially incorrect; usable only with major edits.

- **3 — Mostly Helpful:** Identifies target/violation with minor inaccuracies; usable with light editing.

- **4 — Highly Helpful:** Fully accurate and specific; clearly identifies entities and violations; directly usable as a moderation rationale.

Annotators are additionally given the gold human rationale for reference, consistent with the LLM-as-a-judge setup.

We obtain average helpfulness scores of 1.6 for GRPO and 2.2 for ExPO-HM. After normalizing the scores to the same 0–10 scale used by the LLM-as-a-judge, we observe high agreement between human and LLM evaluations in the relative improvement from GRPO to ExPO-HM. We provide a detailed comparison in Table 11. Inter-annotator agreement is strong, with Krippendorff's $\alpha_{\text{ordinal}} = 0.71$.

Table 11: Comparison of human and LLM evaluation of explanation quality. Human scores are reported in both the original 0–4 Likert scale and a normalized 0–10 scale for comparability with LLM-as-a-judge scores.

| Model | LLM (GPT-4o mini) | LLM (GPT-5) | Human (0–4, Raw) | Human (0–10, Scaled) |
|---|---|---|---|---|
| GRPO | 5.2 | 4.1 | 1.6 | 4.1 |
| ExPO-HM | 6.2 | 5.0 | 2.2 | 5.5 |

# G    CDE DERIVATION AND ALTERNATIVE ESTIMATOR

## G.1    CDE DERIVATION

Here, we provide the full derivation for the CDE metrics estimation. We consider the CDE metrics for a model parameter with $\theta$ over a dataset $\mathcal{D}$.

$$H(d \mid \mathbf{e}, \mathbf{x}) = \mathbb{E}_{\mathbf{x} \sim \mathcal{D}, \, \mathbf{e} \sim \pi_\theta(\mathbf{x}), \, d \sim \pi_\theta(\mathbf{e}, \mathbf{x})} \left[ - \log p_\theta(d \mid \mathbf{e}, \mathbf{x}) \right]$$

By Monte Carlo over the dataset

$$\approx -\frac{1}{|\mathcal{D}|} \sum_{\mathbf{x} \in \mathcal{D}} \mathbb{E}_{\mathbf{e} \sim \pi_\theta(\mathbf{x}), \, d \sim \pi_\theta(\mathbf{e}, \mathbf{x})} \left[ \log p_\theta(d \mid \mathbf{e}, \mathbf{x}) \right]$$

By Monte Carlo sampling $K$ times

$$= -\frac{1}{|\mathcal{D}|} \sum_{\mathbf{x} \in \mathcal{D}} \sum_{i=1}^{K} p_\theta(\mathbf{e}_i \mid \mathbf{x}) \sum_d p_\theta(d \mid \mathbf{e}_i, \mathbf{x}) \log p_\theta(d \mid \mathbf{e}_i, \mathbf{x})$$

Approximate $p_\theta(\mathbf{e}_i \mid \mathbf{x}) \approx \frac{1}{K}$

$$\approx -\frac{1}{K|\mathcal{D}|} \sum_{\mathbf{x} \in \mathcal{D}} \sum_{i=1}^{K} \underbrace{\sum_d p_\theta(d \mid \mathbf{e}_i, \mathbf{x}) \log p_\theta(d \mid \mathbf{e}_i, \mathbf{x})}_{H(d \mid \mathbf{e}_i, \mathbf{x})} \tag{12}$$

For the entropy $H(d \mid \mathbf{e}_i, \mathbf{x})$, we by default compute it over the full decision vocabulary:

$$H(d \mid \mathbf{e}_i, \mathbf{x}) = -\sum_{d \in \mathcal{V}} p_\theta(d \mid \mathbf{e}_i, \mathbf{x}) \log p_\theta(d \mid \mathbf{e}_i, \mathbf{x}), \tag{13}$$

where $\mathcal{V}$ denotes the output vocabulary. For practical efficiency, we do not compute entropy over the entire vocabulary; instead, we approximate it using the top 10–50 tokens by likelihood, which substantially reduces computation and memory costs. When a fine-grained class is represented by multiple tokens, we compute the average token entropy similar to the policy entropy computation.

For binary classification, one may collapse the vocabulary into Yes, No by grouping all tokens semantically aligned with "yes/positive" or "no/negative," and normalizing their probabilities.

## G.2    ALTERNATIVE ESTIMATOR THROUGH CHAIN RULE

When considering the CDE, we can expand through: By the chain rule of entropy:

$$\underbrace{H((\mathbf{e}, d) \mid \mathbf{x})}_{(1)} = \underbrace{H(\mathbf{e} \mid \mathbf{x})}_{(2)} + \underbrace{H(d \mid \mathbf{e}, \mathbf{x})}_{(3)} \tag{14}$$

1. $H((\mathbf{e}, d) \mid \mathbf{x})$ Sequence entropy: the total entropy of generating both reasoning and decision.
2. $H(\mathbf{e} \mid \mathbf{x})$ Reasoning entropy: measures the diversity of reasoning paths the model can produce for an input.
3. $H(d \mid \mathbf{e}, \mathbf{x})$ Conditional decision entropy (CDE): quantifies the uncertainty of the model's decision given its own on-policy reasoning path $\mathbf{e}$.

Both (1) and (2) can be estimated directly via sequence-level sampling. Then CDE, (3) can be obtained by subtraction, using the chain rule in Eq. 14.

## G.3    ALTERNATIVE ESTIMATOR WHEN LOGITS IS NOT AVAILABLE

When model logits are not accessible, we approximate entropy by sampling $K = 16$ responses directly from the LMM and measuring entropy over the final detection decisions.

$$H(d \mid \mathbf{x}) = -\sum_{d \in \texttt{Yes,No}} p_\theta(d \mid \mathbf{x}), \log p_\theta(d \mid \mathbf{x}), \tag{15}$$

where $p_\theta(d \mid \mathbf{x})$ is estimated by counting the relative frequencies of positive vs. negative decisions among the $K$ sampled responses. This can similarly be applied towards fine-grained classes.

To approximate full CDE without logits, one can fix a reasoning trajectory and resample $K$ responses $\mathbf{y}_{ik}$ conditioned on that reasoning (e.g., by sampling with temperature 0.7–1.0). The Monte Carlo estimator in Eq. 12 is then applied to obtain the CDE for each sampled reasoning path. Finally, averaging across multiple such sampled reasonings provides an overall CDE estimate.

## H   CDE FOR CONFIDENCE CALIBRATION

Maximizing the CDE reward implicitly encourages better confidence calibration. By design, the CDE reward penalizes confident wrong predictions and rewards confident correct decisions. This encourages a well-behaved decision distribution: the model becomes confident only when it is likely to be correct, and becomes uncertain when the outcome is ambiguous.

### H.1   CALIBRATION EVALUATION

To validate whether CDE reward improves calibration, we compute Expected Calibration Error (ECE) and Brier score using the model's probability assigned to the final answer token, conditioned on the generated explanation under the Explain-then-Detect setup. For ECE, we use 10 bins.

We evaluate calibration for zero-shot, SFT, GRPO, and ExPO-HM across both 3B and 7B model sizes. As shown in Table 12, ExPO-HM consistently achieves substantially better calibration than all baselines by a large margin. Notably, for the 3B model, ExPO-HM reduces the Brier score from $0.590 \rightarrow 0.283$, indicating significantly improved decision reliability.

Table 12: Calibration metrics (ECE and Brier score) under the Explain–then–Detect setting. Lower values indicate better calibration. ExPO-HM achieves the best calibration across all configurations.

| Model | Variant | ECE ↓ | Brier ↓ |
|---|---|---|---|
| Qwen2.5-VL-3B | Zero-shot | 0.525 | 0.590 |
| | SFT | 0.486 | 0.534 |
| | GRPO | 0.394 | 0.441 |
| | ExPO-HM | **0.232** | **0.283** |
| Qwen2.5-VL-7B | Zero-shot | 0.301 | 0.335 |
| | SFT | 0.234 | 0.282 |
| | GRPO | 0.221 | 0.287 |
| | ExPO-HM | **0.160** | **0.214** |

### H.2   THEORETICAL ANALYSIS

We show that, under the ideal ExPO-HM policy, i.e., when the CDE reward is maximized, the induced decision distribution admits a Brier score that is upper bounded by a small constant determined only by the CDE entropy thresholds $a$ and $b$. This provides theoretical justification for why optimizing CDE improves calibration.

**Setup.**   Following the notation introduced in Sec. 3.1, let $d_i^* \in \{0, 1\}$ denote the ground-truth label. Under our Explain-then-Detect formulation, the model produces a probability

$$p_i' = \pi_\theta(d_i = 1 \mid \mathbf{x}_i, \mathbf{e}_i). \tag{16}$$

To simplify the derivation, we instead let

$$p_i = \pi_\theta(d_i^* \mid \mathbf{x}_i, \mathbf{e}_i), \tag{17}$$

which denotes the model probability assigned to the **correct** class.

The binary Brier score for sample $i$ can then be expressed as

$$\text{BS}_i = (p_i - 1)^2, \tag{18}$$

For convenience, we define correctness by

$$t_i = \begin{cases} 1, & p_i \geq 0.5, \\ 0, & p_i < 0.5. \end{cases} \tag{19}$$

We compute the binary entropy via

$$h_i = H_2(p_i) = -p_i \log p_i - (1 - p_i) \log(1 - p_i). \tag{20}$$

where $H_2$ denotes binary entropy in bits.

**Entropy thresholds imposed by CDE.** The CDE reward uses two entropy thresholds: Low-entropy cutoff $a = 0.1$ and High-entropy cutoff $b = 0.5$, defining:

$$h_i \leq a \quad \text{(high confidence)}, \qquad h_i \geq b \quad \text{(low confidence)}. \tag{21}$$

Let the corresponding confidence thresholds be

$$p_a = H_2^{-1}(a), \qquad p_b = H_2^{-1}(b). \tag{22}$$

For brevity, we write $H_2^{-1}$ for the inverse of $H_2$ on the relevant monotone branch (either $[0, 1/2]$ or $[1/2, 1]$), depending on whether we are in the low- or high-confidence regime.

**Optimal ExPO-HM policy under CDE.** Maximizing CDE encourages:

- *high confidence* ($h_i \leq a$) for correct predictions ($t_i = 1$);
- *high uncertainty* ($h_i \geq b$) for incorrect predictions ($t_i = 0$).

The optimal CDE-maximizing decision distribution satisfies

$$t_i = 1 \Rightarrow 0.5 \leq p_i \leq 1.0 \quad \text{and} \quad p_i \geq p_a, \tag{23}$$
$$t_i = 0 \Rightarrow 0 \leq p_i < 0.5 \quad \text{and} \quad p_i \geq p_b. \tag{24}$$

Thus, we obtain

$$p_i \in \begin{cases} [p_a, 1], & \text{if } t_i = 1 \quad \text{(correct prediction)}, \\ [p_b, 0.5), & \text{if } t_i = 0 \quad \text{(incorrect prediction)}. \end{cases} \tag{25}$$

It is easy to find $p_a = 0.987, p_b = 0.110$ from Eq. 22.

**Bounding the Brier score.** We bound the Brier score by considering the correct and incorrect predictions separately using the probability constraints in Eq. 25.

**Case 1: Correct predictions.** When $t_i = 1$,

$$\text{BS}_i = (p_i - 1)^2 \leq (p_a - 1)^2. \tag{26}$$

**Case 2: Incorrect predictions.** When $t_i = 0$, similarly

$$\text{BS}_i = (p_i - 1)^2 \leq (p_b - 1)^2. \tag{27}$$

Thus, for all samples,

$$\text{BS}_i \leq \begin{cases} (1 - p_a)^2, & t_i = 1, \\ (1 - p_b)^2, & t_i = 0. \end{cases} \tag{28}$$

Table 13: Fine-grained *attack types* results on HatefulMemes. We report overall attack accuracy, micro F1, and per-class F1 for each attack type. Due to space constraints, we use abbreviated labels for attack types; the full names are provided in Table 5.

| Model | Overall Metrics (%) | | Attack Types F1 (%) | | | | | | | |
|---|---|---|---|---|---|---|---|---|---|---|
| | Acc. | Micro F1 | Benign | Dehum. | Infer. | Mock. | Incit. | Excl. | Contempt | Slurs |
| Zero-shot | 44.6 | 44.7 | 72.3 | 36.6 | 14.0 | 16.4 | 19.1 | 37.5 | 5.4 | 16.0 |
| SFT | 57.8 | 58.4 | 78.7 | 43.6 | 5.6 | 12.2 | 18.6 | 22.2 | 9.5 | 22.2 |
| DPO | 63.2 | 63.2 | 81.2 | 44.3 | 22.6 | 26.1 | 20.0 | 19.5 | 6.8 | 21.1 |
| GRPO | 60.4 | 61.2 | 79.6 | 36.5 | 12.0 | 18.4 | 23.4 | 21.7 | 5.4 | 22.2 |
| **ExPO-HM** | **74.8** | **75.6** | **84.6** | **62.1** | **46.0** | **60.0** | **59.5** | **61.2** | **53.3** | **66.7** |

Table 14: Fine-grained *Protected categories* results on HatefulMemes. We report overall accuracy, micro F1, and per-class F1 for each protected category.

| Model | Overall Metrics (%) | | Protected Categories F1 (%) | | | | | |
|---|---|---|---|---|---|---|---|---|
| | Acc. | Micro F1 | Benign | Nationality | Religion | Race | Sex | Disability |
| Zero-shot | 64.3 | 64.5 | 77.7 | 26.7 | 29.2 | 42.5 | 15.4 | 23.5 |
| SFT | 67.6 | 69.4 | 79.9 | 33.4 | 34.4 | 44.7 | 22.6 | 40.0 |
| DPO | 66.1 | 66.6 | 79.1 | 21.4 | 36.6 | 40.8 | 15.4 | 26.7 |
| GRPO | 64.1 | 64.5 | 75.5 | 23.1 | 64.0 | 49.1 | 54.9 | 35.7 |
| **ExPO-HM** | **76.5** | **77.2** | **84.7** | **61.1** | **67.1** | **51.5** | **58.8** | **51.9** |

**Dataset-level bound.** Let $N_{\mathrm{corr}}$ and $N_{\mathrm{wrong}}$ be the number of correct and incorrect predictions in a dataset of size $N$. We have

$$\mathrm{BS} = \frac{1}{N}\sum_{i=1}^{N}\mathrm{BS}_i \le \frac{N_{\mathrm{corr}}}{N}(1 - p_a)^2 + \frac{N_{\mathrm{wrong}}}{N}(1 - p_b)^2. \tag{29}$$

In the extreme (and likely unrealistic) worst case where the model makes errors on all examples ($N_{\mathrm{wrong}} = N$), the bound simplifies to

$$\mathrm{BS} \le (1 - p_b)^2 \approx 0.79. \tag{30}$$

For a balanced binary classification scenario with 50% accuracy ($N_{\mathrm{corr}} = N_{\mathrm{wrong}} = N/2$), the bound becomes

$$\mathrm{BS} \le \frac{(1 - p_a)^2 + (1 - p_b)^2}{2} \approx 0.40. \tag{31}$$

These results show that, under the ideal ExPO-HM policy induced by maximizing the CDE reward, the decision distribution is constrained to a region with substantially lower Brier loss compared to an unconstrained policy. This provides theoretical support that ExPO-HM encourages more calibrated probability estimates.

## I   FINE-GRAINED RESULTS

In addition to the overall micro F1 reported in the main text, we provide detailed fine-grained results on the HatefulMemes dataset for both attack types and protected categories using the Qwen2.5-VL-7B models in Table 13 and Table 14. For both fine-grained setups, we observe that the benign class achieves relatively high performance, whereas most fine-grained categories remain highly challenging for existing baselines: many per-class F1 scores fall below 50%. Recognizing different attack types is particularly difficult for all baselines, and certain protected categories, such as sex and nationality, are especially difficult. Overall, ExPO-HM shows substantial improvements across all fine-grained categories, demonstrating strong gains over all baseline systems.

# J COMPARING GRPO AND ExPO-HM PREDICTIONS

## J.1 CASE ANALYSIS

Table 15 and Table 16 show examples where ExPO-HM corrects prediction errors made by the GRPO baseline on Qwen2.5-VL-7B, illustrating its ability to generate higher-quality explanations and leading to correct final prediction.

Table 15: Part 1 of the Visualization of cases from GRPO Qwen2.5-VL-7B and ExPO-HM Qwen2.5VL-7B Models.

| | Case 1 | Case 2 | Case 3 |
|---|---|---|---|
| Meme |  |  |  |
| Ground Truth | #Hateful | #Hateful | #Hateful |
| GRPO Prediction | #Benign | #Benign | #Benign |
| ExPO-HM Prediction | #Hateful | #Hateful | #Hateful |
| GRPO Explanation | The meme is a humorous observation about a sandwich maker that can be used as an ironing board. It does not contain any offensive or derogatory content directed at any specific group or individual. The joke is based on a practical use of a kitchen appliance and does not express any form of hate or discrimination. | The meme uses a visual pun by showing dogs and labeling them as "a group of men." This is a common internet meme format that uses animals to represent a group of people, often for humorous effect. The image itself does not contain any explicit content or language that could be considered hateful. The use of animals to represent a group of people is a common meme trope and is generally not intended to be offensive. | The meme uses a wheelchair as a visual metaphor for the "hardest part to eat in a vegetable." This is a play on words and is not intended to be offensive. The joke is based on a common phrase and the visual of a wheelchair, which is a tool for mobility, not a derogatory symbol. The meme is not targeting any specific group or race, and the humor is based on a general observation rather than any harmful stereotype. |
| ExPO-HM Explanation | The meme makes a derogatory comparison by using a woman's body to represent an ironing board, which is a dehumanizing and objectifying portrayal. This type of humor is often used to degrade women and is considered dehumanizing. | The meme uses a derogatory term to refer to a group of men, which is a form of dehumanizing and can be considered degrading. The use of a dog to represent a group of men is also an example of speciesism, which is a form of prejudice against non-human animals. While the meme is not explicitly violent, it does contain elements that can be seen as dehumanizing and degrading | The meme uses a derogatory term "vegetable" to refer to a person in a wheelchair, which is a form of dehumanizing and derogatory language. This type of language is considered hateful as it dehumanizes and degrades the person in the wheelchair |

## J.2 ERROR ANALYSIS

During the analysis of the common error cases shared by both the GRPO baseline and ExPO-HM, i.e., cases that ExPO-HM is still unable to correct, we observe two major categories of failures. The first category consists of highly implicit memes, whose harmful intent can only be uncovered through complex, multi-step reasoning. The second category includes memes that require external world knowledge and/or subtle contextual linkage, making their interpretation dependent on background information not explicitly contained in the meme itself. These examples are inherently difficult, and even human annotators may misinterpret them Kiela et al. (2020). Table 17 presents three representative cases.

The first meme uses an apparently nonsensical statement ("potato is my favorite color of the alphabet") to implicitly portray the individual as extremely unintelligent. The models fail to (i) infer the underlying stereotype associated with intellectual disability, and (ii) connect the negative depiction of cognitive ability to the corresponding protected group defined in the policy manual.

The second meme requires linking the reference to 9/11 with broader immigration discourse, yet the models fail to associate the event with anti-immigrant sentiment or identify the targeted group.

For the third meme, the models' rationales focus narrowly on the absence of explicit slurs or group identifiers and interpret "trash" as a generic political metaphor, thereby overlooking its dehumanizing use toward the group depicted in the image.

For future work, developing a larger corpus of implicit hate-speech examples with high-quality chain-of-thought annotations may help pretrain models to better understand subtle and context-dependent hateful content that requires complex reasoning. In addition, incorporating retrieval augmented generation pipeline to supply relevant external knowledge, such as historical, geopolitical, or stereotype-related context, may further improve the model's ability to identify implicit hate that is not directly observable from the meme alone.

Table 16: Part 2 of the Visualization of cases from GRPO Qwen2.5-VL-7B and ExPO-HM Qwen2.5VL-7B Models.

| | Case 4 | Case 5 | Case 6 |
|---|---|---|---|
| Meme |  |  |  |
| Ground Truth | #Hateful | #Hateful | #Hateful |
| GRPO Prediction | #Benign | #Benign | #Benign |
| ExPO-HM Prediction | #Hateful | #Hateful | #Hateful |
| GRPO Explanation | The meme depicts a person covered in soap suds, which is a common reaction to too much soap in a dishwasher. The text humorously suggests that someone has added too much soap to the dishwasher again. There are no elements in the image or text that indicate any form of discrimination or hate towards a specific group. The meme is a light-hearted joke about a common household issue. | The meme makes a claim about the existence of Muhammad based on a scientific discovery, which is a religious and historical topic. However, the meme does not use any offensive language or imagery that would be considered hate speech. It is a humorous and speculative statement, not a direct attack on any group or individual. | The meme appears to be a humorous take on a political situation, using exaggerated expressions and a juxtaposition of images to convey a message. The top image shows a person with a bearded face, possibly representing a political figure. The bottom image shows a person aiming a shotgun, which could be interpreted as a metaphor for taking back what was claimed. The use of such imagery and text is not inherently hateful but rather a form of political commentary or satire. |
| ExPO-HM Explanation | The meme uses a derogatory term "dishwasher" in a way that could be seen as dehumanizing, as it reduces a person to a machine. This type of language can be considered dehumanizing and is often used in a derogatory manner. | The meme makes a false and absurd claim about the existence of Muhammad based on a fictional scenario involving human DNA in a goat carcass. This is not a real scientific discovery and is intended to be humorous at the expense of religious beliefs. The content is not respectful to any religious group and can be seen as mocking or degrading. | The meme contains a provocative statement and imagery that could be interpreted as threatening or aggressive towards muslims. The use of a religious figure and the phrase "we're taking it back" suggests a sense of reclaiming or asserting dominance, which can be seen as a form of hate speech. |

Table 17: Visualization of error cases from GRPO Qwen2.5-VL-7B and ExPO-HM Qwen2.5VL-7B Models.

| | Error Case 1 | Error Case 2 | Error Case 3 |
|---|---|---|---|
| Meme | potato is my favorite color of the alphabet | just under 17 and a half years ago, 19 people were allowed into the us who should not have been / all these years later we are told that the answer to safety is to allow everyone in | so much trash / we need a garbage collector |
| Ground Truth | #Hateful | #Hateful | #Hateful |
| GRPO Prediction | #Benign | #Benign | #Benign |
| ExPO-HM Prediction | #Benign | #Benign | #Benign |
| GRPO Explanation | The meme uses a play on words where "potato" is a pun on the letter "P" in the alphabet. It is a humorous and light-hearted joke that does not contain any offensive or harmful content. | The meme references a tragic event and makes a statement about the safety of allowing everyone into the US. It does not use any offensive language or imagery that would be considered hateful towards a specific group. The statement is a general observation about the balance between safety and freedom of entry, which is a topic of public debate and concern. The meme does not target any particular group or individual in a derogatory manner. | The meme does not contain any explicit hate speech or derogatory language towards a specific group. The content is a political commentary using a metaphor about trash and a garbage collector, which is a common way to express dissatisfaction with a group or situation. The meme does not target any particular race, religion, nationality, or other protected characteristics. |
| ExPO-HM Explanation | The meme does not contain any explicit or implicit content that is considered hateful. It is a humorous statement about a person's preference for the letter 'P' in the alphabet, which is a common internet meme. | The meme does not contain any explicit hate speech or slurs. It is a factual statement about the 9/11 attacks and the subsequent security measures. The language used is neutral and does not express any bias or prejudice against any group. The meme is not intended to incite hatred or discrimination. | The meme appears to be a political statement, likely referencing the attire of the individuals in the image, which is all white. The meme does not contain any explicit or implicit hate speech directed at any specific group or individual. The context and intent of the meme seem to be political rather than hateful. |

