# OpenReview forum: "ExPO-HM: Learning to Explain-then-Detect for Hateful Meme Detection"
_ICLR.cc/2026/Conference — ICLR 2026 Poster_

### Official Review · Reviewer_nfaS · 2025-10-30

**Soundness:** 3
**Presentation:** 3
**Contribution:** 3
**Rating:** 6
**Confidence:** 3

**Summary:**

The paper proposes ExPO-HM, an explain-then-detect framework for hateful meme detection. It addresses two challenges in prior explain-then-detect systems: missing policy-relevant cues in explanations and weak binary rewards that do not guide reasoning. Among three datasets (HatefulMemes, MAMI, PrideMM), ExPO-HM reports state-of-the-art performance on classification.

**Strengths:**

1. The change from direct detection to explain-then-detect in hateful memes is valuable for real-world moderation: explanations provide context, policy linkage, and operational guidance.
2. Extensive experiments are conducted on three public datasets to demonstrate its performance.

**Weaknesses:**

1. The proposed method shows heavy reliance on LLM-as-judge for rationale quality. It's reasonable to introduce different LLMs to evaluate the effectiveness of the proposed three components. Do we need to make new policies for different platforms?
2. Failure cases are not reported.
3. The CDE reward depends on the model’s own decision distribution; while penalizing confident errors helps, more experiments on calibration and robustness would strengthen confidence.
4. Limited discussion on explanation faithfulness. Does the rationale actually reflect features used for the decision?

**Questions:**

Do you have any human evaluations of rationale faithfulness beyond LLM-as-judge?

---

> ### Author Response · Authors · 2025-11-21
> **Response Part 1**
>
> Thank you for your insightful review. In response to your questions:
>
> > The proposed method shows heavy reliance on LLM-as-judge for rationale quality. It's reasonable to introduce different LLMs to evaluate the effectiveness of the proposed three components.
>
> Thank you for raising this concern. In addition to the GPT-4o mini judge (used to ensure a fair comparison with prior work [1]),  we conduct additional experiments with GPT-5 (gpt-5-2025-08-07) and three additional open-source LLM judges, including Qwen3-4B-Instruct-2507, gpt-oss-20b, and gemma-3-4b-it. We also include BERTScore [1], which measures sequence-level semantic similarity between the human-annotated rationale and the model-generated explanation.
>
> | Model         | LLM (GPT-4o mini)↑ | LLM (GPT-5) ↑ | LLM (Qwen3) ↑ | LLM (Gemma-3) ↑ | LLM (gpt-oss) ↑ | BERTScore ↑ |
> | ------------- | ------------------ | ------------- | ------------- | --------------- | --------------- | ----------- |
> | Qwen2.5-VL-3B |                    |               |               |                 |                 |             |
> | Zero-shot     | 3.3  | 1.5           | 2.0           | 2.9             | 2.4             | 0.52        |
> | SFT           | 3.6  | 2.0           | 2.4           | 3.8             | 2.6             | 0.53        |
> | DPO           | 3.5   | 2.2           | 2.3           | 3.7             | 3.0             | 0.53        |
> | GRPO          | 3.8  | 2.8           | 3.0           | 4.2             | 3.2             | 0.53        |
> | ExPO-HM       | 5.1  | 3.6           | 4.2           | 5.5             | 4.0             | 0.56        |
> | Qwen2.5-VL-7B |       |     |      |                 |                 |             |
> | Zero-shot     | 5.0                | 3.8           | 4.5           | 5.2             | 4.0             | 0.55        |
> | SFT           | 5.0                | 4.0           | 4.6           | 5.5             | 3.9             | 0.55        |
> | DPO           | 4.9                | 3.5           | 4.5           | 4.9             | 3.6             | 0.54        |
> | GRPO          | 5.2                | 4.1           | 4.7           | 5.9             | 4.6             | 0.59        |
> | ExPO-HM       | 6.2                | 5.0           | 5.5           | 7.0             | 5.3             | 0.65        |
>
>
>
>
> We note that different LLM judges exhibit different scoring distributions. BERTScore is less sensitive to subtle performance differences, likely due to its limited semantic understanding compared to modern LLMs. GPT-5, Qwen3, and gpt-oss tend to be more strict, while Gemma-3 tends to be more generous. Nevertheless, ExPO-HM models are consistently rated as the best-performing system under all evaluation metrics with substantial margins over GRPO models. This further validates the effectiveness of ExPO-HM.
>
> All results, along with detailed evaluation settings, are provided in Appendix E of the revised paper to ensure full reproducibility.
>
> >Do we need to make new policies for different platforms?
>
> In reality, different platforms do have different hate-speech policies due to regulatory requirements and cultural differences. In our experiments, we evaluate on three datasets: HatefulMemes, MAMI, and PrideMM, which each adopt different hate-speech definitions. Our results show that the proposed method performs well across all three settings.
>
>
> >Failure cases are not reported.
>
> We have added representative failure cases and analysis in Appendix J.2 of the revised paper.

---

> ### Author Response · Authors · 2025-11-21
> **Response Part 2**
>
> >The CDE reward depends on the model’s own decision distribution; while penalizing confident errors helps, more experiments on calibration and robustness would strengthen confidence.
>
> Thank you for raising this question on calibration.
>
> **Connection between CDE and calibration.**
>
> We note that maximizing the CDE reward implicitly encourages better calibration. By design, the CDE reward penalizes confident wrong predictions and rewards confident correct decisions. This encourages a well-behaved decision distribution: the model is confident only when it is likely to be correct, and more uncertain when the outcome is ambiguous.
>
> **Calibration evaluation.**
>
> To validate this, we compute Expected Calibration Error (ECE) and Brier score using the model’s probability outputs of the answer token that is conditioned on the generated explanation under the Explain-then-detect settings. We evaluate calibration for zero-shot, SFT, GRPO, and ExPO-HM across both 3B and 7B model sizes. As shown below, ExPO-HM consistently achieves substantially better calibration than all baselines by a large margin. Notably, for the 3B model, ExPO-HM reduces the Brier score from 0.590 → 0.283, indicating significantly improved decision calibration.
>
> | **Model**     | **Variant** | **ECE ↓** | **Brier ↓** |
> | ----------------------- | ----------------- | --------- | ----------- |
> | Qwen2.5-VL-3B | Zero-shot         | 0.525         | 0.590          |
> |                         | SFT               | 0.486         | 0.534           |
> |                         | GRPO              | 0.394         | 0.441           |
> |                         | ExPO-HM   | 0.232         | 0.283          |
> | Qwen2.5-VL-7B | Zero-shot         | 0.301         | 0.335           |
> |                         | SFT               | 0.234         | 0.282           |
> |                         | GRPO              |    0.221     |    0.287       |
> |                         | ExPO-HM  | 0.160         | 0.214           |
>
> **Theoretical analysis.**
>
> In Appendix H of the revised paper, we further discuss the theoretical upper bound on the Brier score under the optimal ExPO-HM policy (i.e., when the CDE reward is perfectly maximized), illustrating why ExPO-HM optimization leads to better calibrated decision distributions.

---

> ### Author Response · Authors · 2025-11-21
> **Response Part 3**
>
> >Limited discussion on explanation faithfulness. Does the rationale actually reflect features used for the decision? Do you have any human evaluations of rationale faithfulness beyond LLM-as-judge?
>
> Thank you for raising this question regarding explanation faithfulness. To assess the rationale faithfulness beyond LLM-as-a-judge, we conduct two complementary human evaluations on the Hatred corpus (HatefulMemes “dev_seen” set). Each example is independently rated by three crowd-sourced annotators, all with at least an undergraduate degree and demonstrated familiarity with internet meme culture. We evaluate both the GRPO baseline and our ExPO-HM model, and we randomize the order in which model outputs are presented to mitigate ordering or anchoring biases.
>
> (1) Coherence Evaluation
>
> Annotators are asked to judge whether the model’s final decision is logically supported by its rationale. Specifically, for each example, they answer:
>
>
> *“Does the model’s final decision logically follow from its rationale? In other words, is the decision grounded in the explanation provided?”*
>
> Annotators answer Yes / No.
>
> For the GRPO baseline, 96% of the explanations were coherent with the final decision, whereas ExPO-HM achieved 100% coherence.
>
> (2) Helpfulness Evaluation
>
> Annotators also rate how helpful the rationale is for understanding why the meme is hateful. For each example, they answer:
>
>
>  *“How helpful is the model’s rationale for understanding why the meme is hateful or benign?”*
>
> We adopt a 0–4 Likert scale, adapted from prior work [2], defined as follows:
>
> 0 — Unrelated / Incorrect
>  Rationale is irrelevant, incorrect, or unusable; provides no moderation value.
>
>
> 1 — Weakly Related
>  Touches on the meme’s content but lacks specificity or clarity; cannot support moderation.
>
>
> 2 — Partially Helpful
>  Contains some relevant entities or violation cues but is incomplete or partially incorrect; usable only after major editing.
>
>
> 3 — Mostly Helpful
>  Generally correct identification of target/violation with minor inaccuracies; usable with light editing.
>
>
> 4 — Highly Helpful
>  Fully accurate, specific, and concise; correctly identifies entities and violations with no major errors; directly usable as a moderation rationale.
>
> Annotators are also given the gold human rationale as reference, consistent with the LLM-as-judge setup.
>
> We obtain average helpfulness scores of 1.6 for GRPO and 2.2 for ExPO-HM.
> After normalizing the scores to the same 0–10 scale used by the LLM-as-judge, we observe high agreement between human and LLM evaluations in the relative improvement from GRPO to ExPO-HM.
>
>
> Inter-annotator agreement is strong, with Krippendorff’s α (ordinal) = 0.71.
>
> | Model         | LLM (GPT-4o mini)↑ | LLM (GPT-5) ↑ | Human Eval (0 - 4, Raw) | Human Eval (0-10, Scaled) |
> | ------------- | ------------------ | ------------- | ---------------- | ------------------------------- |
> | Qwen2.5-VL-7B |                    |               |                  |                                 |
> | GRPO          | 5.2                | 4.1           | 1.6              | 4.1                             |
> | ExPO-HM       | 6.2                | 5.0           | 2.2              | 5.5                             |
>
> All the experiment details are included in Appendix F of the revised paper.
>
>
> Thank you once again for your in-depth review. We really appreciate your time and effort put into the review to help strengthen the contribution of our paper.
>
>
>
> [1]: Tianyi Zhang, Varsha Kishore, Felix Wu, Kilian Q. Weinberger, and Yoav Artzi. Bertscore:
> Evaluating text generation with bert. In International Conference on Learning Representations,
> 2020.
>
> [2]: Zhilin Wang, Yi Dong, Jiaqi Zeng, Virginia Adams, Makesh Narsimhan Sreedhar, Daniel Egert, Olivier Delalleau, Jane Scowcroft, Neel Kant, Aidan Swope, and Oleksii Kuchaiev. Help-
> steer: Multi-attribute helpfulness dataset for steerlm. In Kevin Duh, Helena Gomez, and Steven
> Bethard (eds.), Proceedings of the 2024 Conference of the North American Chapter of the As-
> sociation for Computational Linguistics.

---

### Official Review · Reviewer_oNni · 2025-11-01

**Soundness:** 3
**Presentation:** 3
**Contribution:** 3
**Rating:** 8
**Confidence:** 3

**Summary:**

This paper introduces EXPO-HM, a method for fine-tuning multimodal local LMs (LMMs) to classify meme images as hateful or not in an “explain-then-predict“ manner, producing both an explanation and a prediction. The method involves two training stages: (1) a warmup SFT stage (SFT-PM) which tunes the LMM to predict the fine-grained image class using a set moderation guideline as additional; (2) a GRPO with curriculum learning (GRPO-CL) stage using a reward function that punishes (A) departure from an explain-then-predict output format and (B) inaccuracy with respect to the ground truth label. The “curriculum“ aspect pertains to the label used--the GRPO-CL process starts out using fine-grained ground truth labels before transitioning to a mix of fine-grained and binary labels. Finally, a variant of GRPO-CL is explored using conditional decision entropy (CDE) CDE as an additional reward signal, which measures the entropy of the label decision conditioned on the produced explanation.

The proposed method is compared to a number of baselines across three hateful meme datasets, including non-explain-then-predict baselines, previously published models, and explain-then-predict models using SFT, DPO, and standard GRPO. EXPO-HM is found to consistently outperform all baselines and prior models on all three datasets. API based models are not compared due to a high rate of rejection of the hateful meme inputs. An ablation study is performed to assess the marginal impact of each part of the training (SFT-PM, GRPO-CL, and the CDE reward component), which finds that all three provide marginal utility. An additional analysis finds that SFT-PM the most effective of a number of possible SFT warmup stages.

**Strengths:**

- **Important task:** The paper covers an important and not particularly solved task, that of hateful meme detection.

- **Comprehensive baselines:** The paper uses a very comprehensive set of baselines, including both prior work, direct detection baselines, and explain-then-detect baselines using a variety of levels of training including SFT alone, DPO, and standard GRPO. The paper provides a reasonable justification for not including API-based models such as the OpenAI o-series. I can’t think of anything that is missing.

- **Good results:** The proposed method outperforms all baselines and prior models. The improvement is a little incremental over the previous state of the art (+~1% F1), but it is consistent across all three datasets and it uses a very different method (no retrieval augmentation).

- **Good ablation studies:** The ablation studies do a nice job of validating the individual pieces of the EXPO-HM method, which might otherwise seem a little arbitrary and over-engineered.

**Weaknesses:**

**Minor** **spelling mistakes:** “taionale”, “subjectuive“ line186

**Questions:**

No particular questions

---

> ### Author Response · Authors · 2025-11-21
>
> Thank you for your detailed and in-depth review of our work.
>
> We appreciate your positive assessment of our paper, including the recognition of the importance of the task we are tackling, the comprehensiveness of our baseline comparisons, the strong performance of EXPO-HM relative to prior methods, and the clarity of our ablation studies. Thank you for noting the minor spelling mistakes. We have corrected these in the revised version and highlighted in blue.
>
> We are grateful for your encouraging feedback and recommendation.

---

### Official Review · Reviewer_ttL2 · 2025-11-01

**Soundness:** 3
**Presentation:** 3
**Contribution:** 3
**Rating:** 6
**Confidence:** 4

**Summary:**

This study propose a method, which is referred to as Explain-then-Detect Policy Optimization, for hateful meme classification and explanation generation, which is mainly utilising the idea of how manual annotation processes are conducted. It demonstrate that combining SFT-warmup, GRPO with curriculum learning and conditional decision entropy improves the performance for both detection and reasoning.

**Strengths:**

- Focused on the limitation of current CoT based approaches, which underperforms SFT based model.
- The proposed approach inspired by how "human annotators are trained for the annotation" is well motivated.
- Experimental results on different meme datasets to show the effectiveness of the proposed approach.
- Task addressed - binary, fine-grained and reasoning
- Ablation studies highlights the importances of the different components of the proposed  approach.
- To ensure reproducibility authors plans to release augmented data and code upon publication.

**Weaknesses:**

- While LLM-as-a judge for explanation/reasoning evaluation is becoming widely used approach, however, they are limited in many setups. Related to this, prompt plays a role for judge and reproducibility if often challenging.
-  Hyperparameters (e.g., CDE) can be sensitive to the dataset, how are they set for different datasets?

Typos/Grammatical issues
- L130: Specify what CXHXW refers to
- L187: scarce taionale corpora
- 213: "study policy guidelines" -> study annotation guidelines?
- 236: "reward In" please check, there might be fullstop or colon.

**Questions:**

- Is it possible to provide details fine-grained results for attack types and targets?
- How exactly were the policy manuals built from the annotation guideline? For examples there is not detailed annotation guideline for hateful memes.

---

> ### Author Response · Authors · 2025-11-21
> **Response Part 1**
>
> Thank you so much for your careful and in-depth review. In response to your questions:
>
> >While LLM-as-a judge for explanation/reasoning evaluation is becoming widely used approach, however, they are limited in many setups. Related to this, prompt plays a role for judge and reproducibility if often challenging.
>
> Thank you for this question. We agree that LLM-as-a-judge evaluation can pose reproducibility challenges despite its widespread use. To ensure reproducibility and fair comparison, we use the same judge model (gpt-4o-mini-2024-07-18) and the same evaluation prompt as prior work [1].
>
> We acknowledge that evaluation with a commercial closed-source model may raise reproducibility concerns, particularly if the model is deprecated in the future. To address this, we conduct additional evaluations using open-source LLM judges with the same prompt for reproducibility, including Qwen3-4B-Instruct-2507, gpt-oss-20b, and gemma-3-4b-it. We also include BERTScore [2], which measures sequence-level semantic similarity between the human-annotated rationale and the model-generated explanation. Because BERTScore does not rely on prompts, it provides a complementary and fully reproducible assessment of explanation quality. We further report GPT-5 (gpt-5-2025-08-07) as a newer and stronger judge for reference.
>
> | Model         | LLM (GPT-4o mini)↑ | LLM (GPT-5) ↑ | LLM (Qwen3) ↑ | LLM (Gemma-3) ↑ | LLM (gpt-oss) ↑ | BERTScore ↑ |
> | ------------- | ------------------ | ------------- | ------------- | --------------- | --------------- | ----------- |
> | Qwen2.5-VL-3B |                    |               |               |                 |                 |             |
> | Zero-shot     | 3.3           | 1.5         | 2.0           | 2.9             | 2.4             | 0.52        |
> | SFT           | 3.6           | 2.0           | 2.4           | 3.8             | 2.6             | 0.53        |
> | DPO           | 3.5          | 2.2           | 2.3           | 3.7             | 3.0             | 0.53        |
> | GRPO          | 3.8                | 2.8           | 3.0           | 4.2             | 3.2             | 0.53        |
> | ExPO-HM       | 5.1                | 3.6           | 4.2           | 5.5             | 4.0             | 0.56        |
> | Qwen2.5-VL-7B |                    |               |               |                 |                 |             |
> | Zero-shot     | 5.0                | 3.8           | 4.5           | 5.2             | 4.0             | 0.55        |
> | SFT           | 5.0                | 4.0           | 4.6           | 5.5             | 3.9             | 0.55        |
> | DPO           | 4.9                | 3.5           | 4.5           | 4.9             | 3.6             | 0.54        |
> | GRPO          | 5.2                | 4.1           | 4.7           | 5.9             | 4.6             | 0.59        |
> | ExPO-HM       | 6.2                | 5.0           | 5.5           | 7.0             | 5.3             | 0.65        |
>
> We note that different LLM judges exhibit different scoring distributions. BERTScore is less sensitive to subtle performance differences, likely due to its limited semantic understanding compared to modern LLMs. GPT-5, Qwen3, and gpt-oss tend to be more strict, while Gemma-3 tends to be more generous. Nevertheless, ExPO-HM models are consistently rated as the best-performing system under all evaluation metrics with substantial margins over GRPO models. This further validates the effectiveness of ExPO-HM.
>
> To further evaluate prompt sensitivity, we manually paraphrased the evaluation prompt and re-evaluated the models using GPT-4o mini. We include the prompt in Appendix E.2 of the revised paper. Results show no significant change, suggesting that our evaluation is not overly sensitive to prompt phrasing.
>
> | Model         | LLM (GPT-4o mini)↑ | LLM (GPT-4o mini, +paraphrase)↑ |
> | ------------- | ------------------ | ------------------------------- |
> | Qwen2.5-VL-3B |                    |                                 |
> | Zero-shot     | 3.3                | 3.2                             |
> | SFT           | 3.6                | 3.4                             |
> | DPO           | 3.5                | 3.3                             |
> | GRPO          | 3.8                | 3.9                             |
> | ExPO-HM       | 5.1                | 5.1                             |
> | Qwen2.5-VL-7B |                    |                                 |
> | Zero-shot     | 5.0                | 5.0                             |
> | SFT           | 5.0                | 5.2                             |
> | DPO           | 4.9                | 5.0                             |
> | GRPO          | 5.2                | 5.1                             |
> | ExPO-HM       | 6.2                | 6.3                             |
>
> All results, along with detailed evaluation settings and prompts, are provided in Appendix E of the revised paper to ensure full reproducibility.

---

> ### Author Response · Authors · 2025-11-21
> **Response Part 2**
>
> >Hyperparameters (e.g., CDE) can be sensitive to the dataset, how are they set for different datasets?
>
> For all GRPO experiments across all datasets, we keep the training hyperparameters fixed and follow the default settings from VeRL, ensuring fairness and reproducibility.
>
> For CDE hyperparameters, we conduct hyperparameter tuning only on the HatefulMemes validation set. After identifying the optimal values, we apply them directly to MAMI and PrideMM, where they also yield strong performance.
>
> Furthermore, as detailed in Appendix C.3 of the revised paper, the CDE hyperparameters have a limited impact on final performance as long as they remain within a reasonable range. This analysis suggests that the selected values generalize well across datasets without requiring per-dataset tuning.
>
> >Is it possible to provide details fine-grained results for attack types and targets?
>
> We report the detailed fine-grained results for attack types and targets in the revised Appendix I.
>
>
> >How exactly were the policy manuals built from the annotation guideline? For examples there is not detailed annotation guideline for hateful memes.
>
> Thank you for this important question regarding our policy manual construction process.
>
> For Facebook HatefulMemes, MAMI, and PrideMM, the dataset authors provide detailed annotation guidelines in prose, including lists of protected or offensive categories (e.g., ethnicity, race, violence) and their definitions. For HatefulMemes specifically, the detailed definitions are released in [4]. We extract this information and convert it into a concise bullet-point list, which we refer to as the policy manual (shown in Appendix B). Below is an example of the original guideline and the derived policy manual.
>
> **Annotation Guideline** [3]:
>
> *"A direct or indirect attack on people based on characteristics, including ethnicity, race, nationality ... We define attack as violent or dehumanizing (comparing people to non-human things, e.g. animals) speech, statements of inferiority…"*
>
> **Policy Manual**:
> - Dehumanizing: Presenting a group as subhuman, explicitly or implicitly
> - Inferiority: Claiming that a group is inferior, less worthy, or less important
> - …
>
> Representing the annotation guidelines in this structured Policy Manual form makes it easier to create targeted instruction-following SFT data compared to using long-form prose descriptions. The conversion from guideline to policy manual is a one-time process performed by a human expert.
>
> We have added a description of this curation process to Appendix B.2 in the revised paper for clarity.
>
> Regarding the typos and grammatical issues, we have corrected them in the revised paper and highlighted all changes in blue.
>
> Thank you once again for your time and effort spent carefully reviewing our work. Your feedback has helped us improve the quality of the paper.
>
>
> [1]: Jingbiao Mei, Jinghong Chen, Guangyu Yang, Weizhe Lin, and Bill Byrne. Robust adaptation of large multimodal models for retrieval augmented hateful meme detection. In Proceedings of the 2025 Conference on Empirical Methods in Natural Language Processing, pp. 23817–23839.
>
>
> [2]: Tianyi Zhang, Varsha Kishore, Felix Wu, Kilian Q. Weinberger, and Yoav Artzi. Bertscore: Evaluating text generation with bert. In International Conference on Learning Representations, 2020.
>
> [3]: Kiela, Douwe, Hamed Firooz, Aravind Mohan, Vedanuj Goswami, Amanpreet Singh, Pratik Ringshia, and Davide Testuggine. 2020. “The Hateful Memes Challenge: Detecting Hate Speech in Multimodal Memes.” Pp. 2611–24 in Advances in Neural Information Processing Systems. Vol. 33.
>
>
> [4]: Lambert Mathias, Shaoliang Nie, Aida Mostafazadeh Davani, Douwe Kiela, Vinodkumar Prabhakaran, Bertie Vidgen, and Zeerak Waseem. Findings of the woah 5 shared task on fine grained hateful memes detection. Proceedings of the 5th Workshop on Online Abuse and Harms (WOAH 2021).

---

### Author Response · Authors · 2025-12-03

Dear Chairs and Reviewers,

We sincerely thank you for the thoughtful and constructive feedback. Here is a summary of the review outcome for your convenience.

## Overall Outcome

All reviewers assessed the submission as a strong and technically sound contribution, with overall ratings of 6, 8, and 6. We carefully addressed all concerns during the author-response period and have incorporated the corresponding updates in the revised manuscript.
## Strengths Highlighted by Reviewers:
1. ExPO-HM addresses an important problem in hateful meme detection (Reviewer oNni), and the shift from direct detection to explain-then-detect provides clear practical value for real-world content moderation (Reviewer nfaS).
2. ExPO-HM addresses a comprehensive set of tasks in hateful meme detection, including binary classification, fine-grained classification, and explanation generation (Reviewer ttL2).
3. ExPO-HM is well-motivated, drawing inspiration from human-annotator training (Reviewr ttL2).
4. All three reviewers recognized that ExPO-HM achieves state-of-the-art results across multiple datasets.
5. All three reviewers noted the comprehensive experimental evaluation, with Reviewers ttL2 and oNni further highlighting the strength of the ablation studies.

## Weaknesses Raised and Our Revisions

### LLM-as-judge reliability and reproducibility.
- To address Reviewer ttL2’s concern about reproducibility and Reviewer nfaS’s request for multiple judges, we added evaluations using four additional LLM judges, including three recent open-source models. Results show consistent trends across all judges.
- To address Reviewer ttL2’s concern about the dependency of the prompt used for the LLM judge, we also tested a paraphrased judge prompt, which produced nearly identical results.
- See Response Part 1 to Reviewer ttL2, Response Part 1 to Reviewer nfaS, and revised Appendix E.
### Explanation faithfulness and Human evaluation.
- In response to Reviewer nfaS, we conducted two complementary human evaluations (coherence and helpfulness). We found high alignment between human assessments and LLM judge scores.
- See Response Part 3 to Reviewer nfaS and revised Sec. 4.7.
### CDE and Model Calibration.
- Addressing Reviewer nfaS’s concern about calibration, we added empirical calibration analysis (ECE, Brier score) and provided a theoretical justification explaining why CDE leads to improved calibration.
- See Response Part 2 to Reviewer nfaS and revised Sec. 4.6.
### Additional analyses and results.
- Detailed CDE hyperparameter analysis (Response Part 2 to Reviewer ttL2; Appendix C.3)
- Fine-grained attack and target-group results (Response Part 2 to Reviewer ttL2; Appendix I)
- A clarified policy-manual construction process (Response Part 2 to Reviewer ttL2; Appendix B.2)
- Expanded case analysis, including representative failure cases (Response Part 1 to Reviewer nfaS; Appendix J.2)


Finally, we thank the area chairs and reviewers once again for their time, thoughtful feedback, and constructive suggestions. We hope our work contributes meaningfully to reducing harmful content and advancing research in this area.

---

### Meta-Review · Area_Chair_V9Wz · 2026-01-07

**Summary:**

This paper proposes ExPO-HM, an explain-then-detect framework for hateful meme detection that is well motivated by human annotation practices. The work addresses key limitations of prior explain-then-detect approaches, namely missing policy-relevant cues and weak reward signals, by combining SFT warmup, curriculum-based GRPO, and a new Conditional Decision Entropy (CDE) objective. Reviewers consistently found the contribution to be sound, clearly presented, and empirically strong. The method achieves state-of-the-art performance across multiple benchmarks on binary detection, fine-grained classification, and explanation quality, supported by thorough ablations and analyses.

**Reviewer Concerns:**

The main concerns raised by reviewers focused on (i) reliance on LLM-as-judge evaluation, (ii) calibration and robustness of the CDE reward, (iii) explanation faithfulness, and (iv) reproducibility details (e.g., prompts and hyperparameters). These concerns were convincingly addressed in the rebuttal through additional experiments with multiple open-source and closed-source judges, prompt sensitivity analysis, human evaluations of coherence and helpfulness, calibration metrics (ECE and Brier score) with theoretical justification, expanded fine-grained results, clarified policy-manual construction, and corrected presentation issues. No substantive concerns remain outstanding.

**Reviewer Scores:**

Reviewer ttL2: Likely improved (6 → 7), with concerns addressed and clarity improved.

Reviewer oNni: Remains strongly positive (8 → 8).

Reviewer nfaS: Likely unchanged (6 → 6).

---

### Decision · Program_Chairs · 2026-01-26

Accept (Poster)